



# Continuous in situ measurements of anchor ice formation, growth and release

Tadros R. Ghobrial[1], Mark R. Loewen[2]

[1]Department of Civil and Water Engineering, Laval University, Quebec, G1V 0A6, Canada
[2]Department of Civil and Environmental Engineering, University of Alberta, Edmonton, T6G 1H9, Canada

*Correspondence to*: Tadros R. Ghobrial (tadros.ghobrial@gci.ulaval.ca)

**Abstract.** In northern rivers, turbulent water becomes supercooled (i.e. cooled to slightly below 0 °C) when exposed to freezing air temperatures. In supercooled water, frazil (small ice disks) crystals are generated in the water column and anchor ice starts to form on the bed. Two anchor ice formation mechanisms have been reported in the literature: either by the accumulation of

suspended frazil particles, which are adhesive (sticky) in nature, on the river bed; or by in situ growth of ice crystals on the bed material. Once anchor ice has formed on the bed, the accumulation typically continues to grow (either due to further frazil accumulation and/or crystal growth) until release occurs due to mechanical (shear force by the flow or buoyancy of the accumulation) or thermal (warming of the water column which weakens the ice-substrate bond) forcing or a combination of the two. Although detailed laboratory experiments have been reported to study anchor ice, but very few field measurements of

anchor ice processes have been reported. These measurements have relied on either sampling anchor ice accumulations from the river bed, or qualitatively describing the observed formation and release. In this study, a custom-built imaging system (camera and lighting) was developed to capture high-resolution digital images of anchor ice formation and release on the river bed. A total of six anchor ice events were successfully captured in the time-lapse images and for the first time, the different initiation, growth and release mechanisms were measured in the field. Four stages of the anchor ice cycle were identified,

namely: Stage 1: initiation by in situ crystal growth, Stage 2: transitional phase, Stage 3: linear growth, and Stage 4: release phase. Anchor ice initiation due to in situ growth was observed in three events and in the remainder the accumulation appeared to be initiated by frazil deposition. The Stage 1 growth rates ranged from 1.3 to 2.0 cm/hr and the Stage 2 and 3 growth rates varied from 0.3 to 0.9 cm/hr. Anchor ice was observed releasing from the bed in three modes referred to as lifting, shearing and rapid.

## 1 Introduction

Anchor ice is defined as ice that is attached or "anchored" to the bed of natural water bodies (rivers, lakes or sea floors) as defined by World Meteorological Organization (1970). Although observations on the formation of anchor ice in rivers have been documented since the 18[th] century (Barnes, 1908), the mechanisms of formation, growth and release, as well as its overall effect on river ice processes, is still a relatively unstudied phenomenon (Tsang, 1982; Beltaos, 2013). Anchor ice formation




and release can cause significant changes to the river bed geometry, affect water levels and discharges, and consequently loss of hydropower production during freeze-up seasons (e.g. Girling and Groeneveld, 1999; Jasek et al., 2015). Anchor ice released from the bottom often contains significant amounts of bed materials which contributes to the sediment transported in river systems (e.g. Kempema and Ettema, 2011; Kalke et al., 2017). Recently it has been shown that the duration and extent of

anchor ice cycles has an effect on algae growth rates and its total biomass (Suzuki et al., 2018). Also, fish habitat, in particular fish spawning can be affected by the formation of anchor ice on the bed which can block oxygen supply to substrate water and freeze the eggs under the ice (e.g. Prowse, 2001; Brown et al., 2011). Available river ice numerical models have attempted to include the effects of anchor ice in hydraulic modelling. These models have mostly relied on empirical or semi-empirical relations (e.g. Shen, 2010; Lindenschmidt, 2017; Blackburn and She, 2019; Makkonen and Tikanmäki, 2018) but the

development of physically-based models has been challenging due to the lack of accurate field measurements of anchor ice formation, growth and release.

The process of anchor ice formation starts when surface water becomes supercooled (i.e. water is cooled below its freezing point) typically due to freezing air temperatures. In the presence of sufficient flow turbulence, the supercooled surface water

is transported to lower layers and quickly reaches the river bed (Daly, 1994). In supercooled turbulent water, active (sticky) frazil ice crystals are typically generated in the water column and subsequently anchor ice may also form on the river bed. It has been established that anchor ice formation can be initiated by two processes: in situ growth of ice crystals (i.e. nucleation of ice crystals atop the bed material) and/or accretion (i.e. deposition) of active frazil particles on submerged objects (Tsang, 1982). After the initial formation of anchor ice crystals on the river bed, the accumulation continues to grow either by crystal

growth due to heat loss to the surrounding supercooled water, and/or by further deposition of suspended frazil particles (Osterkamp and Gosink, 1983; Qu and Doering, 2007). The final thickness of the anchor ice layer is limited by several factors including the absence of supercooled water, the stream flow depth (although in some cases the accumulation can emerge above the water surface forming anchor ice dams), the growth of an overlaying surface layer of stationary or border ice, or the release of the anchor ice accumulation from the bottom (e.g. Osterkamp and Gosink, 1983; Beltaos, 2013; Turcotte et al., 2013).

Increasing anchor ice thickness coincides with increases in the drag and buoyancy forces acting on the accumulation. Anchor ice release is thought to occur due to mechanical or thermal forcing or a combination of the two (e.g. Parkinson, 1984; Shen, 2005). Mechanical release of anchor ice occurs when the buoyancy and drag forces are greater than the ice-substrate bond, or when these forces are greater than the submerged weight of the anchor ice and the attached bed materials (sands, gravel, or boulders). This latter mechanism results in rafting of river bed materials (sediments) as released anchor ice pans are advected

downstream (Kempema and Ettema, 2011; Kalke et al., 2017). Anchor ice has often been observed to release in the morning following cold and clear nights (Barnes, 1908; Kempema et al., 2001; Daly and Ettema 2006) and this has been attributed to warming of the water by solar radiation that weakens the ice-substrate bond leading to thermal release.



Many of the physical measurements available on anchor ice formation, growth and release, are from detailed laboratory experiments (Kerr et al., 2002; Doering et al., 2001; Qu and Doering, 2007). In laboratories, the ambient conditions are controlled (air temperature, discharge, and channel characteristics) and the environment is favourable for conducting detailed measurements (e.g. video recordings, depth and velocity profiles) for collecting samples of anchor ice accumulations. Kerr et

al. (2002) conducted a laboratory study on anchor ice formation and its hydraulic effects on a gravel bed in a refrigerated flume. In all of their experimental runs, they observed anchor ice initiation only due to frazil deposition/attachment to the bed (i.e. no in situ growth). They documented three distinct stages of anchor ice growth: initial, transitional, and final growth stages. The initial stage (referred to as "Stage 1" herein), is the formation of the first visible anchor ice crystals layers on the substrate. This stage was characterized by faster growth rates and uneven appearance along the bed. The transitional stage

(referred to as "Stage 2" herein) started once the accumulations began to emerge out of the substrate and protruded into the flow. The hydrodynamic drag forces acting on the accumulation caused the flattening or release of the anchor ice formation. During this transitional stage, anchor ice accumulation either continued to increase due to frazil deposition, or decreased slightly due to its release into the flow. The final growth stage (referred to as "Stage 3" herein) was defined as the nearly uniform (linear) and slower growth rates of anchor ice thickness due to continuous frazil deposition. At this stage, individual

anchor ice forms were not distinguishable. The measured growth rates ranged between ~ 0.02 and 0.05 cm/°C-hr, which for an average air temperature of -16.0 °C translates to a growth rate between 0.4 and 0.8 cm/hr. Although they reported that anchor ice released only at Reynolds number less than ~22,000, it is worth noting that they only tested two values of Reynolds numbers (i.e. 21,800 and 29,000).

Doering et al. (2001) and Qu and Doering (2007) conducted laboratory experiments on anchor ice evolution in a counter rotating flume. They measured growth rates between 0.3 and 0.7 cm/hr and their data suggested that anchor ice growth rates and densities increase with the Froude number. Although they visually observed that anchor ice always initiated with frazil deposition, with a careful interpretation of the water temperature (supercooling) curves, they were able to attribute some of the continuous growth of anchor ice thickness to the in situ growth of the crystals. They found that gravel size did not have an

effect on the formation (initiation) mechanisms, but they found that anchor ice releases more easily when it was attached to smaller gravel particles. They also observed that anchor ice tends to release when the Reynolds number is less than 42,000.

Several field observations on anchor ice processes has been reported in the literature using grab sampling techniques, on-shore photographs, and intermittent underwater photography. Therefore, these observations were mostly limited to the qualitative

description of anchor ice properties such as shape, thickness, extent of formation and release, and how these properties are related to hydro-meteorological conditions (e.g. Hirayama et al., 1997; Terada et al., 1998; Turcotte and Morse, 2011; Nafziger et al., 2017). Reasons for this limitation include the need for supercooled water, that anchor ice typically forms during night time when visibility is low, the difficulty of predicting where anchor ice might form, and of course the limitation of working in cold weather. As a result, continuous measurements of anchor ice growth rates and observations of initiation processes (i.e.



in situ crystal growth versus frazil particles deposition) have not been made in the field. Despite these limitations, field studies have advanced our knowledge of anchor ice processes and provided many valuable information.

Hirayama et al. (1997) conducted a three years study on a small gravel stream (6 to 9 m wide and 0.3 to 0.6 m deep) in

Hokkaido, Japan. They mapped the Froude number contours within the study reach and showed that anchor ice only forms when the Froude number is between 0.2 and 1.5. They reported that sampled anchor ice masses consisted mainly of needle-like crystals which were growing either in the upstream or the downstream direction of the flow. They showed that the thickness and volume of anchor ice accumulations increased with the cumulative degree hour of freezing of air temperature, and reported anchor ice thicknesses ranging between 3 and 17 cm. Also, they measured anchor ice accumulations densities between 300

and 700 kg/m$^3$ and showed that the density tended to increase with the flow velocity.

Several studies on the Laramie River, USA (e.g. Kempema and Ettema., 2009; Kempema and Ettema, 2011) reported that daily anchor ice cycles (formation at night and release in the morning) can generate anchor ice accumulation thickness between 0.2 and 0.3 m with no apparent relation between crystal sizes and Froude number. They also observed the morphology of

anchor ice accumulations and reported that large plate-like ice crystals exceeding 10 cm in length were most dominant and that disc-like crystals are rarely seen. They observed that these large crystals result from in situ growth of frazil crystals that become attached to the bed. While studying anchor ice rafting by sediments, they concluded that anchor ice release and associated ice rafting were diurnal events, which suggests that solar radiation was an important factor.

Stickler and Alfredsen (2009) conducted a detailed study on anchor ice formation at three sites: two sites on unregulated rivers (Southwest Brook, Canada, and River Sokna, Norway), and one site on a regulated stream (River Orkla, Norway). They concluded that anchor ice formation is mainly due to frazil deposition and is dependent on the flow turbulence (i.e. Reynolds number) with no apparent correlation with Froude number. They reported that in low turbulence areas (median Reynolds number of ~ 10,000) anchor ice grew in the vertical direction with a soft texture and low densities (between 360 and 600

kg/m$^3$), smaller frazil crystals (<0.01 m), and thinner accumulation thicknesses (median of 0.04 m). In high turbulence areas (median Reynolds number of ~ 25,000), anchor ice grew in both the vertical and lateral direction with much higher densities (between 600 and 900 kg/m$^3$), larger particles (up to 0.1 m) and larger accumulation thicknesses (median 0.07 m). Anchor ice release was only observed when the water temperature increased above zero and when shortwave radiation (direct sunlight) reached the bed.

Dubé et al. (2014) investigated the characteristics of anchor ice accumulations on the Montmorency River, Canada using thin sections analysis and computed axial tomography (CAT) scans of the collected anchor ice samples. Elongated columnar ice crystals were observed only in ice dam samples and disk-shaped ice crystals were observed in both ice dam samples and submerged anchor ice samples. In both cases, the crystals showed grew preferentially perpendicular to the flow. Their results



also suggested that in situ growth of disk-shaped ice crystals was the dominant process for the formation of anchor ice and ice dams. They reported individual ice crystals between 3 and 6 cm in length with mean accumulation porosity between 38 and 44%.

Jasek et al. (2015) and Jasek (2016) monitored anchor ice release in the Peace River, Canada. They showed that anchor ice formation and release caused significant fluctuations in discharge and water levels, which caused ice cover instability and consolidation, and consequently freeze-up jams. Their observations showed that anchor ice release appeared to be mainly due to hydraulic effects, rather than thermal influence of the sun. Acoustic scanning of the river bottom indicated length to thickness ratios of ~ 24:1 for the anchor ice patches on the bed. They estimated ice coverage to be ~ 70% for a reach of over 200 km in

length.

Nafziger et al. (2017) studied three streams in New Brunswick, Canada (2 unregulated and 1 regulated) and 161 anchor ice formation/release events were observed using time-lapse photographs from the shore. A correlation was found between the increase in water depth (stage) during the formation of an anchor ice event, and the corresponding accumulated freezing degree

hours of air temperature. Although there were no direct measurements of growth of anchor ice accumulation, the "trend" of the increase in water levels showed good agreement with the laboratory growth rates reported by Kerr et al. (2002). On days with a net heat gain at the water surface and air temperatures > -15°C, 98 % of anchor ice accumulations completely released, which indicates a strong thermal control on anchor ice release.

As discussed above, previous field and laboratory studies have provided considerable insight into the formation, release, and properties of anchor ice but there are still considerable gaps in our knowledge. For example, the relative importance of frazil deposition versus in situ growth, mechanical versus thermal release and single versus multi-day cycles. However, one of the most critical gaps is that the anchor ice growth rates and mechanisms observed in the laboratory have never been confirmed in the field. The primary goal of this study was to address this gap by making direct measurements of anchor ice growth in the

field. For this purpose, a custom-built underwater imaging system (camera and lighting) was deployed on the North Saskatchewan River in Edmonton. The imaging system was able to capture for the first time high-resolution digital time lapse images of anchor ice formation, growth and release mechanisms. This paper describes the deployments of the imaging system, results from the continuous measurements of anchor ice processes, and the effect of ambient hydro-meteorological conditions on these processes.



## 3 Study Site and Methods

### 3.1 Study site

The North Saskatchewan River, Canada, (length ~1300 km; mean annual discharge of 245 m³/s at the downstream end at Prince Albert) is a glacier-fed, regulated river that flows east from the Canadian Rockies (1,800 m above sea level) across

Alberta (720 m above sea level at Edmonton), to central Saskatchewan (Kellerhals et al., 1972). The river reach within Edmonton is irregularly meandering with many point bars and side channel bars. It ranges in depth between 1 and 3 meters, and between 150 to 250 meters in width (Gerard and Andres, 1982). The bed material is composed of glacial till and alluvial sands and gravels (Kellerhals et al., 1972). The winter discharge is largely controlled by the outflows from the Bighorn and Brazeau dams in the upper part of the basin and the average daily winter discharge at Edmonton is 126 m³/s (Hicks, 1997).

During winter, hydropeaking power generation at the two dams typically causes water level fluctuations of 30 to 50 cm at the study site. Freeze-up on the North Saskatchewan River in Edmonton can start as early as mid-October and a complete ice cover can form as late as the end of December. Figure 1 presents a map showing the study site on the North Saskatchewan River in Edmonton. Measurements of anchor ice were conducted at the City of Edmonton Quesnell Bridge (53°30′20″N 113°33′60″W) during freeze-up season (October to December) of 2018. The river in this reach has a slope of ~ 0.0002 and a width of ~ 200

m. This site was selected for its accessibility and because anchor ice has been observed forming in this reach during previous studies (e.g. Kalke et al., 2017).

### 3.2 Instrumentation

In order to capture high-resolution photographs of anchor ice properties in the field, an underwater imaging system and artificial substrate were designed and built as shown in Fig. 2. The imaging system consisted of a 36-megapixel Nikon D800

digital single-lens reflex (DSLR) camera (equipped with a Micro-Nikkor 35 mm f/1.8D lens) coupled with a Nikon SB-910 Speedlight flash. Both the camera and the flash were contained in underwater housings which were mounted side-by-side on a MiniTec aluminum rail. The imaging system was secured in a 100 cm long, 50 cm wide, and 20 cm high PVC frame. The aluminum rail was designed to release from the frame using two 20 cm high handles and a pivot hinge assembly. This feature allowed the rail to be lifted out of the frame so that images could be examined and batteries changed without removing the

entire system from the river. The frame was equipped with two ten-pound weights to help anchor the system to the river bed. In order to prevent frazil and anchor ice from forming on the camera and flash lenses, a 10 m long pipe heat trace cable was wrapped around both underwater housings and then covered with insulating bubble wrap. The heat trace cable was extended using a 30 m power cord laid out along the river bed and connected to a Subaru R1700i gas generator secured on the river bank. The generator needed to be filled with gas every ~7 to 8 hours.

An artificial substrate was constructed and bolted to the imaging system frame as shown in Fig. 2. Although imaging anchor ice formation directly on the natural river bed would be ideal, it is very difficult to pre-adjust the camera settings and lightings



to acquire clear images of the forming anchor ice. However, using a constructed substrate allowed us to conduct preliminary laboratory experiments to adjust these settings in a controlled environment. In addition, a constructed substrate offered the opportunity to observe multiple anchor ice events growing on an identical substrate, eliminating variation in bottom sediment properties as an experimental variable. Lastly, it also allowed us to closely examine and photograph any anchor ice deposits

that had not released once the system was removed from the water. To make the constructed substrate as similar as possible to the natural river bed, bed material was sampled from the river at the deployment site in early October 2018. The bed samples were oven dried and sieved in the University of Alberta geotechnical lab. Gravels/boulders that were greater than 3.8 cm (1.5 inch) in size were used for the substrate. The substrate materials were hand-picked so that they have one relatively flat side and they ranged in size from 3.8 cm to 12.5 cm. The gravel/boulders were then glued to a 50 by 50 cm wide plywood base.

Multiple 2.5 cm diameter holes were drilled into the base to reduce buoyancy forces on the substrate.

Initial imaging settings (including ISO, aperture, focus and duration of the flash pulse) and the distance between the substrate and the lens face were determined in the laboratory by immersing the system in a tank of tap water. These camera settings and the setup configuration were modified over the course of the field deployments to improve image quality and increase the

camera battery life. These modifications included: decreasing the distance from the camera housing window to the in-focus bed material from 60 cm to 40 cm; adding a 25 mm extension tube to the camera lens; changing the underwater camera housing from a clear Ikelite D800 housing to a coated aluminum Aquatica AD800 housing; and increasing the image sampling interval from 30 seconds to 5 minutes. Adding the extension tube and moving the substrate closer resulted in a field of view in the images of 34 by 18 cm as opposed to 45 by 30 cm for the original configuration. Increasing the image sampling interval

extended the camera battery life from about 6 hours to 24 hours.

In addition to photographing anchor ice, the water temperature was measured to investigate the effect of temperature variations on anchor ice formation and release. Measurements were made using two RBR SoloT (accuracy ±0.002°C) temperature loggers sampling every 5 seconds: one mounted on the substrate and another attached to the top of the frame at 20 cm above the bed

(see Figure 2). The anchor ice imaging system was deployed a total of four times during the freeze-up season. During each field deployment, the instruments were carried from the south bank of the river and installed on the river bed. For the first three deployments, the system was installed 15 to 20 m from the south shore (between the right bank and the first Quesnell Bridge pier) where the water depth ranged between 0.6 and 0.7 m at the start of the deployments (see Fig. 1). For the last deployment, the instruments were lowered from the border ice ~ 100 m upstream of the bridge and ~ 30 m from the south

shore in a water depth of 1.6 m.

Meteorological data was downloaded from the Alberta Climate Information Service (ACIS) website. The closest weather station (approximately 2.0 km southeast of the study site, see Fig. 1) was the "Edmonton South Campus UA" station (Climate ID 3012220) which provides hourly weather data for the air temperature, solar radiation, wind speed and direction,



rainfall/snowfall depth and relative humidity. Real-time hydrometric data for North Saskatchewan River at Edmonton was obtained from Water Survey of Canada gauge #05DF001 in a 5-minute interval. The gauge is located approximately 10.7 km downstream of the study site (Fig. 1).

## 4 Data Analysis

A careful examination of the images showed that the imaging system was able to capture individual anchor ice crystals growing on the artificial substrate and also the thickness of anchor ice accumulation. Therefore, the images were processed to primarily estimate these two quantities. All of the captured anchor ice images were processed in two steps. First, the images were enhanced using image processing software (BatchPhoto Pro ®). The enhancements included: stamping the date/time when the image was taken, auto adjusting the contrast, reducing the hue, increasing the saturation, increasing the lightness, and reducing

the noise in the images. These enhancements corrected for the continuous change in ambient lighting and flow turbidity over the course of a single deployment.

Second, the enhanced images were imported into MATLAB ® and the edge of anchor ice crystals and the anchor ice accumulation thickness was manually tracked as a function of time using the image processing toolbox. For the crystal growth

measurements, the images were visually examined to identify a number of crystals (typically between 1 and 4 crystals) that were clearly visible in consecutive images. Then the pixel coordinates of the edge of each identified crystal were manually tracked and extracted from the series of images. The pixel distance between the edge of the same crystal on successive images was scaled by using the in-focus size of the substrate material. The total length of the crystal was then calculated to estimate the growth of the accumulation with time. A processed image showing the individual crystals forming on the substrate is

presented in Fig. 3a. The anchor ice accumulation thickness was measured by manually tracking multiple points across the top of the accumulation in each image. The average accumulation thickness was calculated for each image by averaging these manually tracked points across the width of the image. A processed image showing anchor ice accumulation atop the substrate is shown in Fig. 3b.

## 5 Results

### 5.1 Synopsis of Field Deployments and Anchor Ice Events

Figure 4 presents time series of the air temperature and the river stage measured from 1-Nov-2018 to 31-Dec-2018. The first ice pan was observed in the river on 7-Nov-2018 and the river was completely ice covered on 23-Dec-2018. The freeze-up season lasted almost 46 days and was one of the longest in recent years. During freeze-up, the weather forecast was monitored and the dates of the deployments were determined based on when supercooling of the river was expected to occur and the

availability of the research team. The anchor ice imaging system was deployed a total of four times (referred to as DEP-1 to





DEP-4) during the 2018 freeze-up season as highlighted in Figure 4. Table 1 summarizes the camera settings and the duration and timing of each deployment.

Figures 5 to 8 present time series of the measured air and water temperatures, solar radiation, river stage, and anchor ice thickness during the field deployments DEP-1 to DEP-4, respectively. Prior to deployments DEP-1 and DEP-2, the air temperature was above zero and dropped to -10 and -15°C, respectively during the deployments, and frazil pans more than 30 cm thick were observed passing by the deployment site. For DEP-1 the instruments were deployed at 20:00 on 11-Nov-2018 and were retrieved at 8:00 on 12-Nov-2018; a duration of 12 hrs. The recorded anchor ice event during DEP-1 was labelled as Event A (see Figure 5). During this event, anchor ice started to form on the substrate at 21:00 on 11-Nov-2018 shortly after the deployment started. Unfortunately, the camera stopped working from 1:50 until the battery was replaced at 4:30 on 12-Nov-2018. During this event, the water was continuously supercooled at approximately a constant temperature of -0.009 °C. When the instruments were retrieved at 9:00 on 12-Nov-2018, anchor ice was still attached to the substrate and therefore, no images of anchor ice release were captured during this event. The quality of the images captured during this event did not allow the tracking of individual crystals, but the top of the anchor ice accumulation was visible in the images. Figure 9 presents a photograph of the substrate after Event A showing a significant accumulation of anchor ice during this 12-hour event.

DEP-2 started at 18:00 on 15-Nov-2018 and ended at 12:00 on 17-Nov-2018, lasting for 40 hours (see Fig. 6). Although DEP-2 lasted two nights, anchor ice did not form during the first night because the water was above 0°C. During this event, a classic supercooling curve was observed, which reached a maximum supercooling temperature of -0.09°C at approximately 17:00 on 16-Nov-2018 and then warmed up to an average residual temperature of approximately -0.02°C. Anchor ice crystals started to form on the substrate at ~16:30 on 16-Nov-2018 just before the water reached its maximum supercooling and shortly before the solar radiation reached zero W/m$^2$. The camera stopped working from 2:25 until 5:53 on 17-Nov-2018 when the camera battery was replaced. At about 6:40 the heat trace stopped working because the generator stalled and the lens was completely covered with ice. When the instruments were retrieved at 11:40, some anchor ice was still attached to the substrate but it seems that the majority of the detected anchor ice in the images had released sometime after 6:40 and prior to retrieval. Although this deployment experienced several instrument failures, it was possible to track individual crystal growth and the thickness of anchor ice accumulation in the images.

After DEP-2, the air temperature stayed relatively warm until 1-Dec-2018 when the temperature dropped below zero. DEP-3 started at 16:00 on 3-Dec-2018 when the air temperature decreased from -5 to -15°C and lasted 43 hours until 12:00 on 5-Dec-2018 when the instruments were retrieved (see Figure 7). Events C and D were both captured during DEP-3. During these two events, the camera and the generator did not encounter any issues and worked throughout the entire deployment period. Anchor ice started to form on the substrate at 16:40 and 17:55 shortly after the solar radiation reached zero W/m$^2$ and released the next morning at 8:20 and 7:15 shortly before sunrise at 8:30 for Events C and D, respectively. During this deployment, the water





was constantly supercooled at about -0.009°C except when it decreased to -0.018°C around the time of the release of Event C. Images from Events C and D were used to extract anchor ice crystals growth and thickness of anchor ice accumulation.

After DEP-3, the air temperature gradually warmed again but rafts and ice pans were still observed in the river. On 15-Dec-
2018 the temperature dropped from above 0 to -10°C and the river started to stage-up due to higher surface pan concentrations and possibly multiple bridging locations downstream of the study site. DEP-4 lasted for 49 hours from 16:30 on 15-Dec-2018 until 17:30 on 17-Dec-2018 (see Fig. 8). During this deployment, the water was constantly supercooled at about -0.01°C. Events E and F were both captured during DEP-4. Anchor ice started to form on the substrate at 16:45 and 16:25 (immediately after the solar radiation reached zero W/m$^2$) and released in the afternoon of the next day at 13:20 and 14:40 for Event E and
F, respectively. It was believed that these events lasted longer because the instruments were deployed in a much deeper location (1.6 m deep as opposed to ~ 0.6 m for the previous deployments) which decreased the effect of heating by solar radiation. Due to higher turbidity images from Events E and F were only used to extract the thickness of anchor ice accumulation since it was not possible to distinguish individual crystals in the images.

At the end of each deployment, after the retrieval of the instruments from the river, images of anchor ice that had not released from the substrate were taken (e.g. Fig. 9). From these images, three distinct anchor ice crystal shapes were observed on the substrate as shown in Fig. 10. These shapes are: (a) curved needle crystals that grew on the surface of the bed material from the contact edges between adjacent gravels towards the centre of the gravel from all sides; (b) platelet crystals that grew starting in the interstitial spaces between gravel particles and then grew vertically away from the gravel, typically angled upstream;
and (c) disk shaped crystals that look like typical suspended frazil ice crystals that attached to the substrate.

**5.2 Anchor Ice Formation, Growth and Release**

The processed images from each anchor ice event were combined in time lapse videos to help visualizing the results. An example of such videos for Event C is available for download at https://doi.org/10.7939/DVN/6X5ATL (Ghobrial and Loewen, 2019). Using these videos, the process of anchor ice formation, growth and release was separated into four stages, namely:
Stage 1: initiation by in situ growth, Stage 2: transitional phase, Stage 3: linear growth, and Stage 4: release phase. For illustration purposes, these stages are labelled on the time series results of anchor ice thickness measured during Event C as shown in Fig. 11. The results of all anchor ice events are summarized in Table 2. Initiation of anchor ice by in situ crystal growth (Stage1) was only observed in Events B, C, and D (see Fig. 6c and 7c). During Stage 1 individual anchor ice crystals started to grow off the substrate typically angled in the upstream direction of the flow. It is of interest to examine the rate of
anchor ice crystals growth observed during these events. For this purpose, a total of nine crystals were tracked, four crystals from each of Events B and C, and one crystal from Event D. The growth of the leading edge of these crystals is plotted against time in Fig. 12. Note that the time scale for each crystal length measurements was referenced to when the individual crystal first appeared in an image. The individual crystals grew at an approximately linear rates ranging from 0.9 to 2.4 cm/hr with an





average of 1.7 cm/hr. This stage lasted between ~ 1.5 to 3.0 hours (typically between 18:00 and 21:00). At the end of Stage 1, the crystals ranged between 2.8 cm and 7.7 cm in length.

Stage 2 is a transitional period when individual crystals came in contact with each other and were not easily distinguished in the photographs. During this stage, the surface of the anchor ice started to become flattened by the flow due to the increased drag force and then continued to grow through the deposition of suspended frazil crystals and flocs and/or further interstitial crystal growth. This stage was only distinguishable during the three events B, C, and D when in situ crystal growth was observed. This stage lasted for ~4 hours (typically between 21:00 and 1:00). In Stage 3 the deposition was already flattened out and the anchor ice accumulation had a distinct upper surface which continued to grow upwards at an approximately constant rate due to frazil deposition. This stage lasted for ~8 hours (typically between 1:00 and 7:00). For Events A, E, and F frazil deposition appeared to be the initiation mechanism of anchor ice formation and therefore Stage 1 was not observed during these three events. Consequently, for analysis purposes, the growth of the anchor ice accumulation during Stage 2 and 3 (combined) is plotted in Fig. 13 for all six events. The time scale for each event was referenced to when anchor ice first appeared in the images for Events A, E, and F; but for Events B, C, and D the start time was referenced to when Stage 1 ended. The average rate of accumulation growth during all events ranged between 0.3 and 0.9 cm/hr with an average of ~ 0.6 cm/hr (Table 2). At the end of Stage 3 the anchor ice accumulation thickness due to frazil deposition only was between 5.3 and 9.6 cm.

Stage 4, the release of anchor ice, was recorded for Events C, D, E and F but not for Events A and B due to equipment malfunction or due to retrieval prior to the release. Three modes of anchor ice release were observed in the data: lifting, shearing, and rapid release. During the release of Events C and E, the entire anchor ice accumulation was observed lifting up away from the substrate until it suddenly completely released. This mode lasted ~20 and ~100 min with a lifting rate of 9.0 and 3.3 cm/hr for Events C and E, respectively. During Event D, consecutive layers of accumulation were sheared off with the top layer releasing first followed by the bottom layer. This release mode lasted ~ 20 min. The release of the entire accumulation during Event F occurred in less than 5 minutes (i.e. the time between consecutive images) and therefore no lifting or shearing was observed.

## 6 Discussion

Three stages of anchor ice growth very similar to those reported by Kerr et al. (2002) were observed for the first time in the field in this study using time-lapse photographs. Three of the six anchor ice events (Events B, C, and D) were observed to be initiated by in situ crystal growth (Stage 1) followed by frazil deposition. For the remaining three events (Events A, E, and F) no in situ crystal growth was observed and it appeared that the accumulations grew only by frazil deposition (Stage 2 and 3). It should be noted that Kerr et al. (2002) did not report observing in situ crystal growth and attributed the faster growth in



Stage 1 to only frazil deposition. Qu and Doering (2007) did not directly observe in situ thermal growth in their anchor ice images but did conclude that it occurred in their laboratory experiments based on careful analysis of water temperature time series. Kempema and Ettema (2009) studied anchor ice crystal morphology on a small riffle-and-pool stream and collected anchor ice samples that were comprised of large blade-shaped crystals up to 5 cm in length (see their Fig. 1). They concluded

5 that these larger crystals were formed by in situ growth of suspended frazil crystals that had become attached to the bed. Furthermore, they wrote that anchor ice formed initially by adhesion of frazil ice crystals to the bed and subsequent growth occurred by a combination of frazil accretion and in situ growth. It is unclear if they are referring to the adhesion of a relatively small number of suspended frazil ice crystals to the bed that subsequently acted as nucleation sites for the growth of large crystals. Or if they are referring to the adhesion of a sufficient number of crystals that a layer of measureable thickness is

10 initially formed and that in situ growth occurred within this layer. Their description of anchor ice formation is certainly consistent with events A, E and F in which only frazil deposition was observed. However, for events B, C and D in which in situ growth of large crystals was initially observed there is some uncertainty. Initial adhesion of suspended frazil ice crystals to the bed prior to the start of large crystal growth was not observed in the time-lapse images but this could be because the suspended crystals were much too small to be visible. Therefore it is possible that the first step in anchor ice formation is the

15 adhesion of suspended frazil crystals to the bed and in this case it would follow that some of these crystals could then act as nucleation sites for the large crystals that were observed growing on the bed in this study. This process would also be consistent with Kempema and Ettema's (2009) description of their field observations.

The initial or Stage 1 crystal growth rates measured in this study ranged from 1.3 to 2.0 cm/hr and this is within the range of

20 0.4 to 2.3 cm/hr reported by McFarlane et al. (2016) for growth rates of dendritic frazil ice crystals observed in a sequence of images of a frazil floc trapped between two cross-polarizing filters at the same field site. Initial growth rates of anchor ice accumulations in a laboratory channel were estimated from the slopes of the curves plotted in Fig. 19 from Kerr et al. (1997) and these varied from approximately 1.7 to 2.8 cm/hr. Kempema and Ettema (2013) observed anchor ice growing on wedge wire screens in the Laramie River and plotted crystal growth as a function of time (see their Fig. 8, 9 and 10). The initial

25 growth rates estimated from these plots ranged from approximately 1.0 to 4.0 cm/hr. However, it is uncertain if wedge wire screens accurately model the growth of anchor ice on the river bed. The wedge wire screens were mounted above the bed where velocities would be higher and it is possible that the slightly higher observed growth rates were due to higher turbulent heat transfer rates associated with these elevated water velocities. The field and laboratory measurements of anchor ice growth on the channel beds suggest that initial growth rates range from approximately 1.0 to 3.0 cm/hr.

The time lapse images of anchor ice during Stage 2 and 3 indicate that the growth of the accumulations was mainly due to frazil deposition. It is possible that further in situ crystal growth in the interstitial spaces between the deposited frazil crystals occurred and this would increase the accumulation density and strengthen the bond between crystals. The rate of growth of anchor ice accumulation during Stage 2 and 3 ranged between 0.3 and 0.9 cm/hr. This rate is in agreement with the laboratory



measured rates of 0.4 to 0.8 cm/hr reported by Kerr et al. (2002) and 0.3 to 0.7 cm/hr reported by Doering et al. (2001). It is interesting to note that in this study the average rate of crystal growth (Stage 1) of 1.7 cm/hr was ~ 3 times the average rate of the accumulation growth due to deposition of frazil (Stage 2 and 3). In order to examine the effects of heat loss on the growth rate of anchor ice, following Kerr et al (2002), the thickness of accumulation from all the events during Stage 2 and 3 were

plotted against the cumulative degree hour of freezing air temperature as shown in Fig. 14. The linear rates of growth observed in the field ranged between 0.05 and 0.12 cm/°C-hr compared to 0.02 to 0.05 cm/°C-hr observed in Kerr's experiments. Nafziger et al. (2017) plotted the rise of water level due to anchor ice formation against the cumulative degree hours of freezing and also found that most of the observed events have rates of growth higher than 0.05 /°C-hr which is in agreement with this study.

Total anchor ice thicknesses measured in this study (at the end of Stage 3) ranged between 6.1 and 15.4 cm. This range is consistent with the ranges reported in some previous studies, e.g.: 3 to 17 cm by Hirayama et al. (1997), 20 to 30 cm by Kempema et al. (2001), and 7 to 10 cm by Stickler and Alfredsen (2009). The crystals sizes observed in this study ranged between 2.8 cm and 7.7 cm and are in agreement with previous field studies, e.g.: 3 to 6 cm by Dubé et al. (2014), and up to

10 cm by Kempema and Ettema (2011). However, several studies do report substantially thicker accumulations. Tremblay et al. (2014) reported thicknesses ranging from 0.18 to 0.46 m in a small river (width 6-12 m) and Evans et al. (2017) reported accumulations up to ~0.9 m thick, using side-scan sonar in a much larger river (width 220-440 m). During this study, the research team observed anchor ice accumulations projecting out of the water surface in water depths of 1.6 m or greater at the deployment site. At an average growth rate of 0.6 cm/hr it would take 267 hours for 1.6 m of anchor ice to accumulate. If

anchor ice grows only during ~12 hours a day of supercooling, about 22 days of growth would be required to accumulate 1.6 m. Also, several holes augered through the border ice showed that the bottom of border ice was in contact with and possibly supported on the underlying anchor ice accumulation in water depths up to ~1.5 m. Nafziger et al. (2017) also observed this phenomenon. Anchor ice accumulations this thick are unusual in this reach of the North Saskatchewan River but the freeze-up season was much longer in 2018 than is typical. Ice pans first appeared on the river Nov. 7, 2018 and a solid ice cover was

not formed until Dec. 23, 2018 a total duration of 46 days approximately twice as long as the typical duration. Evidently this much longer freeze-up duration enabled much thicker accumulations of anchor ice to form than is typical.

Four of the six anchor ice events observed in this study started within 0.5 hr of sunset and the remaining two events started 1.6 and 4.3 hr after sunset. This is consistent with what would be expected for diurnal anchor ice events that begin in the late

afternoon or evening. At this time of day the combined effect of decreasing shortwave solar radiation and lower air temperatures typically leads to an increase in the net heat flux from the water to the atmosphere that initiates anchor ice formation. Events C and D occurred in shallow water (~0.6 m) and released the following morning at 8:20 and 7:15 or 0.2 and 1.3 hours prior to sunrise, respectively. Events E and F occurred in deeper water (~1.6 m) and released in the afternoon at 13:20 and 14:40 or 4.5 and 6 hours after sunrise, respectively. In all four cases the water remained supercooled at the observed





residual temperature for the entire duration of each event indicating that there was no warming of the water detected by the temperature logger mounted on the substrate. The fact that release occurred while the water temperature remained constant and supercooled is evidence that release was not caused by the melting of the bond with the substrate due to warming water. However, it might be possible for the bond to melt due to heat transfer from the sediments with no observable change in the

water temperature. Evidence for the role of shortwave solar radiation in the release is inconclusive since two accumulations released prior to sunrise and two near mid-day. The fact that the two accumulations in deeper water released later in the day suggests that solar radiation might have played a role in the release since this effect would take longer to penetrate deeper water but this is not conclusive evidence. The air temperature at the time of release for Events B, C, D and E was -10, -9.7, -5.0 and -0.5°C, respectively. This is in agreement with Nafziger et al. (2017) who reported that most of the anchor ice release

events they observed occurred when there was a positive heat flux to the water and the air was warmer than -15°C.

The possibility that mechanical forces triggered the release was also considered. Buoyancy and hydrodynamic forces always play some role in anchor ice release since they are always present. This can be illustrated by considering two limiting cases. In the first case the ice-substrate bond is weakened by thermal effects and one or both forces lifts or shears the accumulation

off the bed. This would be characterized as thermal release since it was thermal radiation and/or heating that triggered the release. In the second case the strength of the ice-substrate bond remains constant and the magnitude of one or both forces increases triggering release. This would be characterized as mechanical release. In some cases both the strength of the ice-substrate bond and the magnitude of the forces may be varying and then release could be triggered by both a weakening of the bond and an increase in one or both of the forces. In this study the four accumulations (Events C, D, E, F) grew to thicknesses

that ranged from 6.1 to 15.4 cm and then released. Nafziger et al. (2017) estimated the strength of the anchor ice-substrate bond using an equation proposed by Malenchak (2011) and by assuming that the anchor ice accumulation released solely due to buoyancy forces. In order to make similar calculations we assumed the anchor ice density varied from 300-700 kg/m³, ice density was equal 917 kg/m³, substrate diameter ranged from 0.038 to 0.125 m (range of rock sizes used in the constructed substrate) and the accumulation thickness varied from 0.061 to 0.154 m (the range of observed thicknesses). This method gave

estimates of the anchor ice-substrate bond strength that ranged from 18 to 111 N/m² which is comparable to Nafziger et al.'s (2017) values that ranged from 45 to 138 N/m².

It is difficult to quantitatively assess the role of hydrodynamic forces in the release of the four events since the only information available is the approximate local depth and the data from the Water Survey of Canada gauge (#05DF001). The gauge data for

events C and D does not appear to be ice affected with water levels varying from approximately 3.15 to 3.45 m due to diurnal hydropeaking (see Fig. 7b). During events E and F, the water level was steadily rising from 3.16 to 3.71 m and did not follow the typical diurnal pattern indicating that the presence of ice was affecting the gauge (see Fig. 8b). The release of event C anchor coincided with a peak in the daily water levels of 3.38 m. The release of events D occurred during rising water levels 5-6 hours prior to the daily peak and the stage was 3.45 m. During events E and F the water levels were also rising (see Fig.





8b) likely due to a combination of hydropeaking and backwater effects related to ice congestion downstream and release occurred at a stage of 3.36 and 3.70 m, respectively. Therefore, all four anchor ice events that were observed releasing did so when water levels were rising or were approaching the daily maximum. This may indicate that hydrodynamic forces played a role in the release of these anchor ice accumulations, but it is difficult to conclude this with any certainty.

The rate of anchor ice growth is currently calculated in most river ice process models using the following equation,

$$\frac{dh}{dt} = \frac{\gamma C_v}{(1-e_a)} + \frac{\phi_{wi}}{\rho_i L_i (1-e_a)} \tag{1}$$

where $h$ is the anchor ice thickness, $t$ is time, $\gamma$ is the frazil accretion rate to the bed, $C_v$ is the volumetric concentration of

suspended frazil, $e_a$ is the porosity of anchor ice, $\rho_i$ is the density of ice, $L_i$ is the latent heat of ice and $\phi_{wi}$ is the net rate of heat transfer from the ice to the water (Shen et al., 1995). The first term on the right hand side models growth via frazil deposition and the second term models in situ growth and decay. There are two variables ($C_v$ and $\phi_{wi}$) to be predicted each time-step and four parameters ($e_a$, $\rho_i$, $L_i$ and $\gamma$) to be set to constant values in this equation. The density and latent heat of ice are typically assumed to be 917 kg/m$^3$ and 334 kJ/kg, respectively. The porosity of anchor ice samples collected in the field

have ranged from 0.38 to 0.56 (Dubé et al., 2014; Jasek, 2016). The anchor ice samples collected in those two studies were either firm enough to maintain their structural integrity when removed from the water or were taken from released anchor ice pans. Newly formed anchor ice accumulations likely have higher porosities because they often do not maintain their structural integrity when sampling is attempted. The porosity of frazil ice flocs has been estimated to be approximately 0.80 (Schneck et al., 2019) and this may represent a reasonable upper limit for the porosity of newly deposited anchor ice. Values

for the accretion rate $\gamma$ in the literature range from approximately $10^{-6}$ to $10^{-3}$ (Malenchak, 2011) and measurements of the volumetric concentration of frazil $C_v$ reported in the literature vary from $10^{-6}$ to $10^{-2}$ (McFarlane et al., 2019). In this study growth rates due to deposition during Stages 2 and 3 were observed to vary from 0.3 to 0.9 cm/hr. Using the average growth rate of 0.6 cm/hr and assuming the porosity is 0.4, the value of the numerator $\gamma C_v$ is predicted to be $10^{-6}$. This suggests that $\gamma$ is likely not significantly less than ~$10^{-4}$ m/s since this would require that $C_v$ be significantly greater than ~$10^{-2}$ which is not

realistic.

During Stage 1 when in situ growth of anchor ice was observed the crystal growth rates ranged from 1.3 to 2.0 cm/hr. Using these values and porosities of 0.4 and 0.8, the resulting range in $\phi_{wi}$, the net rate of heat transfer from the ice to the water, is estimated to be 220 to 1030 W/m$^2$. The higher heat flux corresponds to lower porosity and higher growth rate. Note that for

the water to remain at a constant temperature, as was observed in this study, the heat flux from the ice to the water must be balanced by an equal net heat flux from the water to the air. The net heat flux from the water to the air can be estimated using a linear heat transfer equation with a heat transfer coefficient of 20 W/m$^2$ °C, a typical value for North American rivers





(Beltaos, 2013). The lowest air temperature during the three anchor ice events when in situ growth was observed was -15°C and therefore the maximum net water-air heat flux would be estimated to be 300 W/m². This suggests that the lower limit of 220 W/m² is a more realistic estimate of the net ice to the water heat flux and therefore that the porosity of anchor ice formed by in situ thermal growth might be closer to 0.8 than 0.4.

**7 Conclusions**

The first direct field measurements of anchor ice growth and release were captured in this study. Three stages of growth similar to those reported by Kerr et al. (2002) were observed in the time-lapse images. A total of six anchor ice events were captured and growth due to frazil deposition and in situ growth was observed in three events and in the remainder only frazil deposition occurred. Anchor ice was observed releasing from the bed in three modes referred to as lifting, shearing and rapid. The Stage

1 growth rates measured by tracking the growth of individual crystals on the substrate ranged from 1.3 to 2.0 cm/hr and these were comparable to rates observed in previous laboratory and field studies. The measured growth rates in Stage 2 and 3 due to frazil deposition varied from 0.3 to 0.9 cm/hr which were comparable to measurements made in two previous laboratory studies. It is worth noting that in this study significantly higher growth rates ranging from 0.05 to 0.12 cm/°C-hr were observed compared to the rates of 0.05 cm/°C-hr or less reported by Kerr et al. (2002).

All of the observed anchor ice accumulations began forming in the afternoon or evening between 16:30 and 21:00. The release of four of the accumulations was captured in the time-lapse images and occurred between 7:15 and 14:40. The two events in shallow water released just prior to sunrise and the two events in deeper water in the early afternoon. There is evidence that solar radiation, buoyancy and hydrodynamic forces may have all played some role in the timing of the releases. It does not

seem likely that release was triggered directly by hydrodynamic forces because the water level and flow rate variations were not significant at the time of release. The fact that during all four events the water temperature remained supercooled at residual temperatures of approximately -0.01°C is clear evidence that weakening of the ice-substrate bond by warming water was not a factor. It seems likely that the two events in shallow water released due to buoyancy since both released prior to sunrise. However, all that can be concluded regarding the release of the two deeper water accumulations is that buoyancy forces and/or

solar radiation may have played a role. Clearly additional research investigating the factors that cause anchor ice release is required.

River ice process models currently use a semi-empirical equation to model anchor ice growth due to frazil deposition and in situ growth. This simple equation accounts for the two observed growth mechanisms and is based on sound physics combined

with reasonable engineering approximations (Shen et al., 1995). Analysis of the term in the equation modelling frazil deposition leads to the conclusion that the frazil accretion rate $\gamma$ must be greater than or equal to ~$10^{-4}$ m/s. Analysis of the second term that models in situ growth suggests that the porosity of newly formed anchor ice may be significantly larger than
0.4, the value that is typically used in model studies. Unfortunately, the field data gathered in this study did not enable an assessment of the accuracy of this equation. A first step in accomplishing this task would be to capture accurate simultaneous measurement of suspended frazil concentrations, anchor ice porosity and anchor ice growth rates under a variety of conditions. These would ideally be field measurements but due to the significant challenges of making these types of measurements in the

field it might only be possible to perform them in the laboratory. This data would allow calculations of the frazil accretion rate and heat flux from the anchor ice by inverting the existing equation. This has the potential to improve application of the existing equation in two ways. First, it could provide an empirical method for specifying the value of the frazil accretion rate as opposed to the current practice which is to set the value based on judgement alone. Secondly, the estimates of the anchor ice to water heat flux could be used to improve the method used to compute it that uses the same equation used to predict the heat flux

between the ice cover and the water. Finally, it is worth noting that recent progress improving and validating river ice process models by comparison to field data has been reported in the literature (Blackburn and She, 2019; Pan et al., 2020; Wazney et al., 2019).

**Data availability.**

Data are available from the authors upon request.

**Author contributions.**

TRG and MRL designed the apparatus and performed the field work together. TRG carried out the analysis and processing of the data, prepared the figures, and wrote the manuscript with review and contributions from MRL.

**Competing interests.**

The authors declare that they have no conflict of interest.

**Acknowledgements.**

We would like to thank the Natural Sciences and Engineering Research Council of Canada (NSERC) for supporting this project (RGPIN-2015-04670 and RGPAS 477890-2015) and Perry Fedun for his valuable technical assistance. We are grateful for that support.

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

5    World Meteorological Organization (WMO): WMO sea-ice nomenclature, codes, and illustrated glossary. Geneva: World Meteorological Organization Rep 259, TP145, 1970.



**Tables**

**Table 1: Summary of field deployments and camera settings.**

| Deployments ID | DEP-1 | DEP-2 | DEP-3 | DEP-4 |
|---|---|---|---|---|
| Average local water depth (m) | 0.6 | 0.6 | 0.7 | 1.6 |
| Camera lens | 35 mm lens | 35 mm lens | 35 mm lens + 25 mm extension tube | 35 mm lens + 25 mm extension tube |
| Underwater camera housing | Ikelite D800 | Ikelite D800 | Ikelite D800 | Aquatica AD800 |
| Distance between lens and substrate (cm) | 60 | 60 | 40 | 40 |
| Field of view (width by height, cm) | 45 by 30 | 45 by 30 | 34 by 18 | 34 by 18 |
| Deployment start date (DD/MM/YY) and time (HH:MM) | 11/11/18 20:14 | 15/11/18 19:27 | 03/12/18 16:26 | 15/12/18 16:44 |
| Deployment end date (DD/MM/YY) and time (HH:MM) | 12/11/18 8:54 | 17/11/18 11:41 | 5/12/18 11:44 | 17/12/18 17:30 |
| Deployment Duration (hr) | 12.7 | 40.2 | 43.3 | 48.8 |



**Table 2: Summary of results for measured anchor ice events.**

| Event ID | Event A | Event B | Event C | Event D | Event E | Event F |
|---|---|---|---|---|---|---|
| Formation date (DD/MM/YY) and time (HH:MM) | 11/11/18 21:00 | 16/11/18 16:30 | 3/12/18 16:40 | 4/12/18 17:55 | 15/12/18 16:45 | 16/12/18 16:25 |
| Release date (DD/MM/YY) and time (HH:MM) | - | - | 4/12/18 8:20 | 5/12/18 7:15 | 16/12/18 13:20 | 17/12/18 14:40 |
| Event duration (hr) | - | - | 15.7 | 13.3 | 20.6 | 22.2 |
| Air Temperature Range ( °C) | -5.6 to -10 | -13 to -15 | -7.4 to -10.5 | -2.5 to -9.7 | -1.7 to -9.9 | -0.2 to -9.2 |
| Water Temperature ( °C) | -0.009 | -0.03 to -0.088 | -0.009 | -0.009 | -0.010 | -0.010 |
| Stage 1 average growth rate (cm/hr) | - | 1.3 | 1.9 | 2.0 | - | - |
| Stage 2&3 average growth rate (cm/hr) | 0.6 | 0.4 | 0.8 | 0.9 | 0.3 | 0.4 |
| Accumulation thickness (cm) | 7.6 | 8.7 | 15.4 | 11.3 | 6.1 | 10.6 |
| Stage 4: Release mechanism | - | - | Lifting | Shearing | Lifting | Rapid |





**Figures**

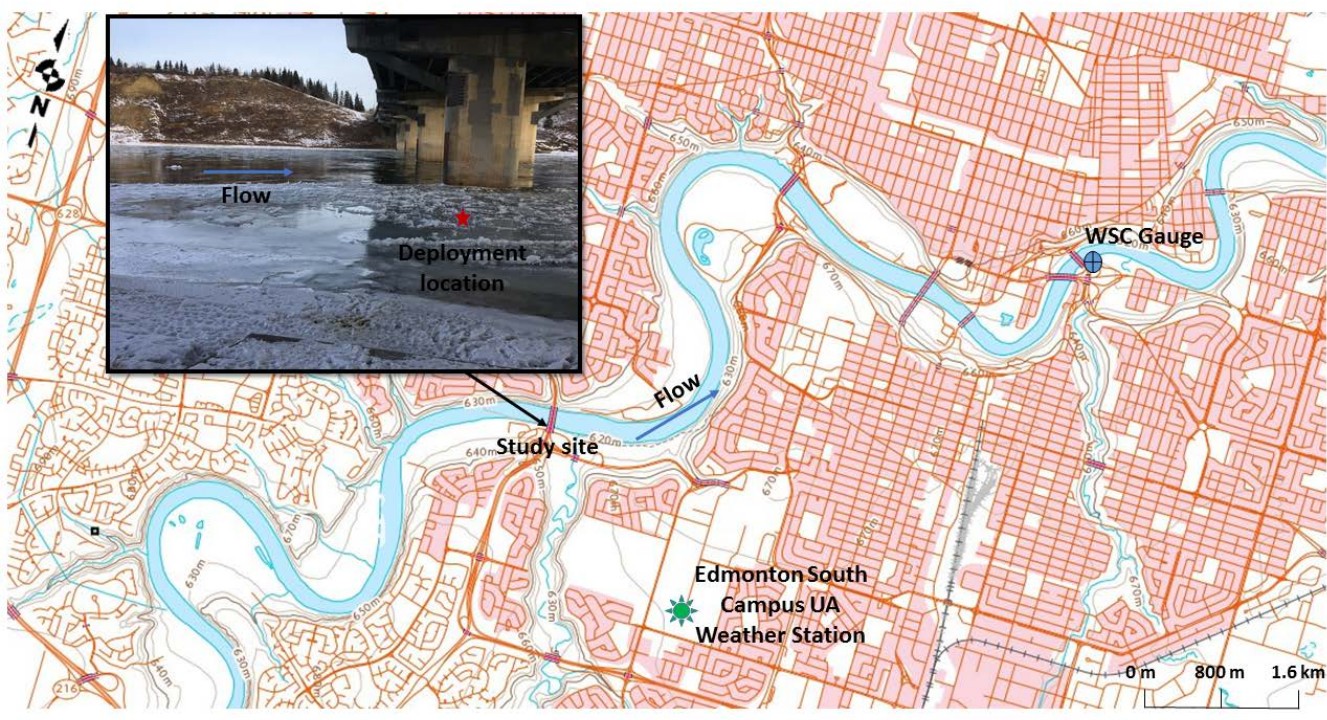

**Figure 1: Map showing the study site on the North Saskatchewan River in Edmonton at the Quesnell Bridge (the base**
5 **map was downloaded from the Atlas of Canada-Toporama website). The inset is a photo from the right bank looking**
**north showing the deployment site under the bridge.**



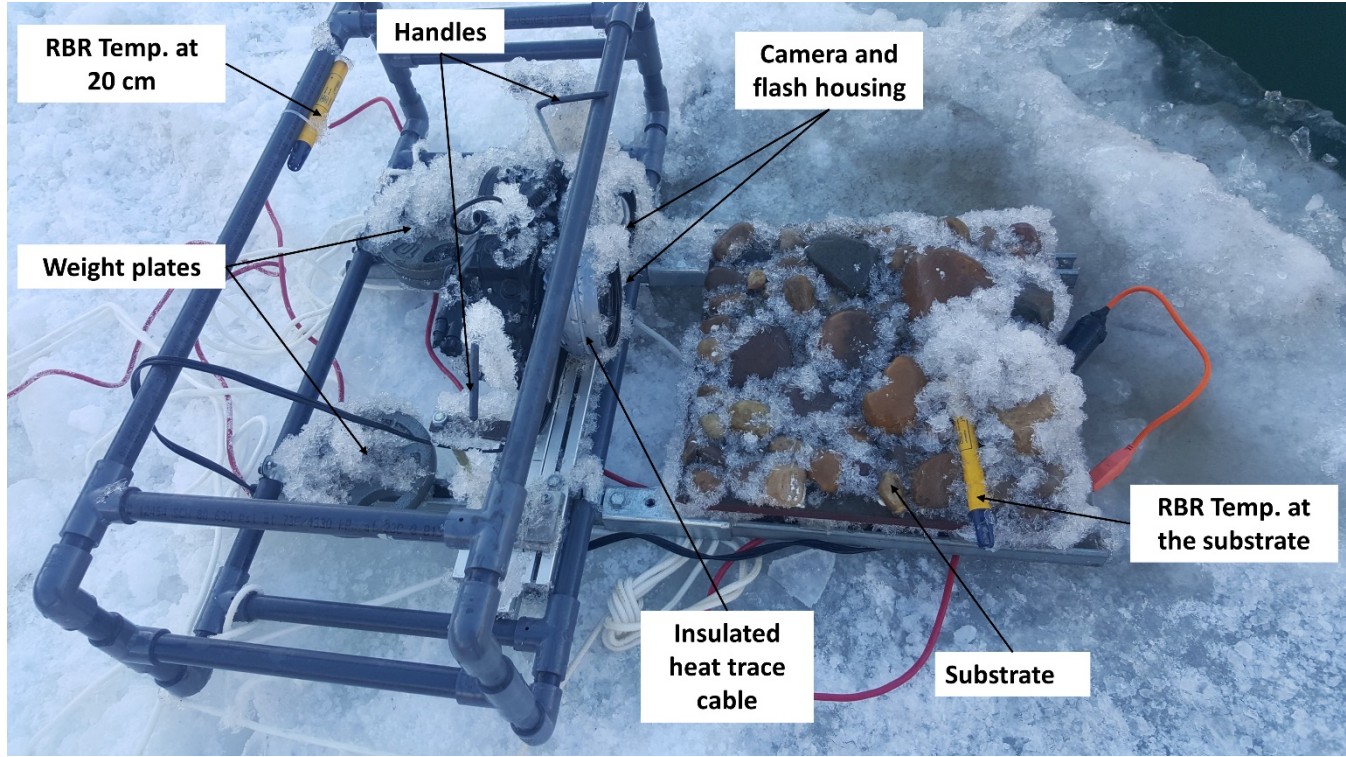

**Figure 2: Anchor ice imaging system and artificial substrate after its retrieval from the river on Nov. 12th, 2018.**



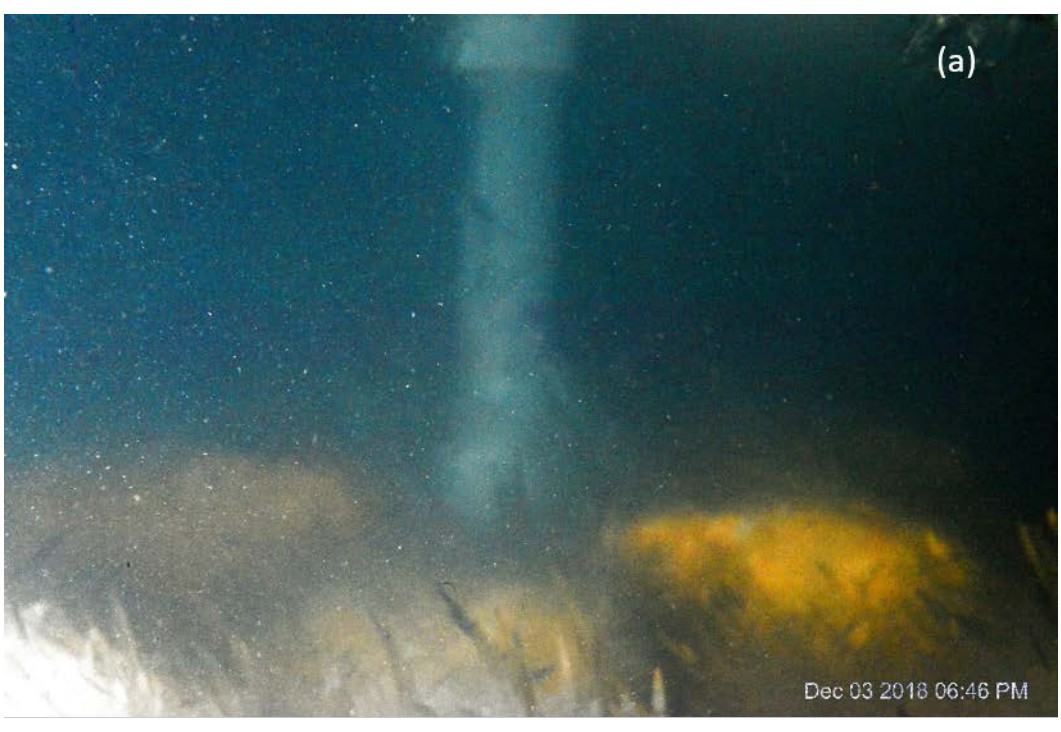

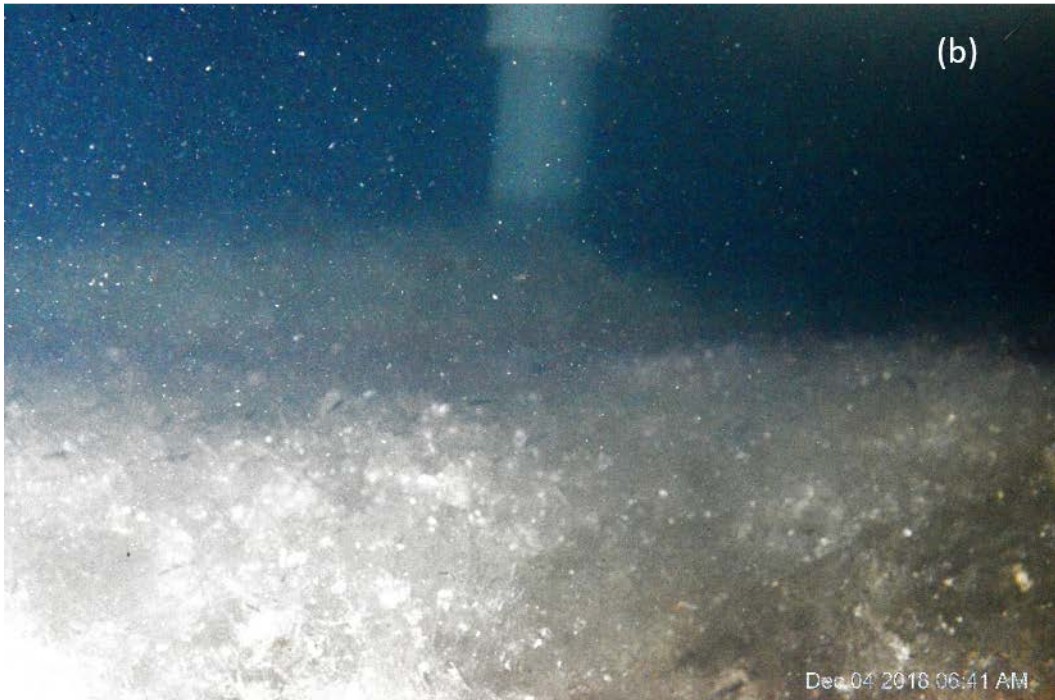

**Figure 3: Sample processed images from an anchor ice event on Dec. 3&4, 2018 (Event C) showing: (a) individual crystals growing off the substrate, and (b) anchor ice accumulation over the substrate.**



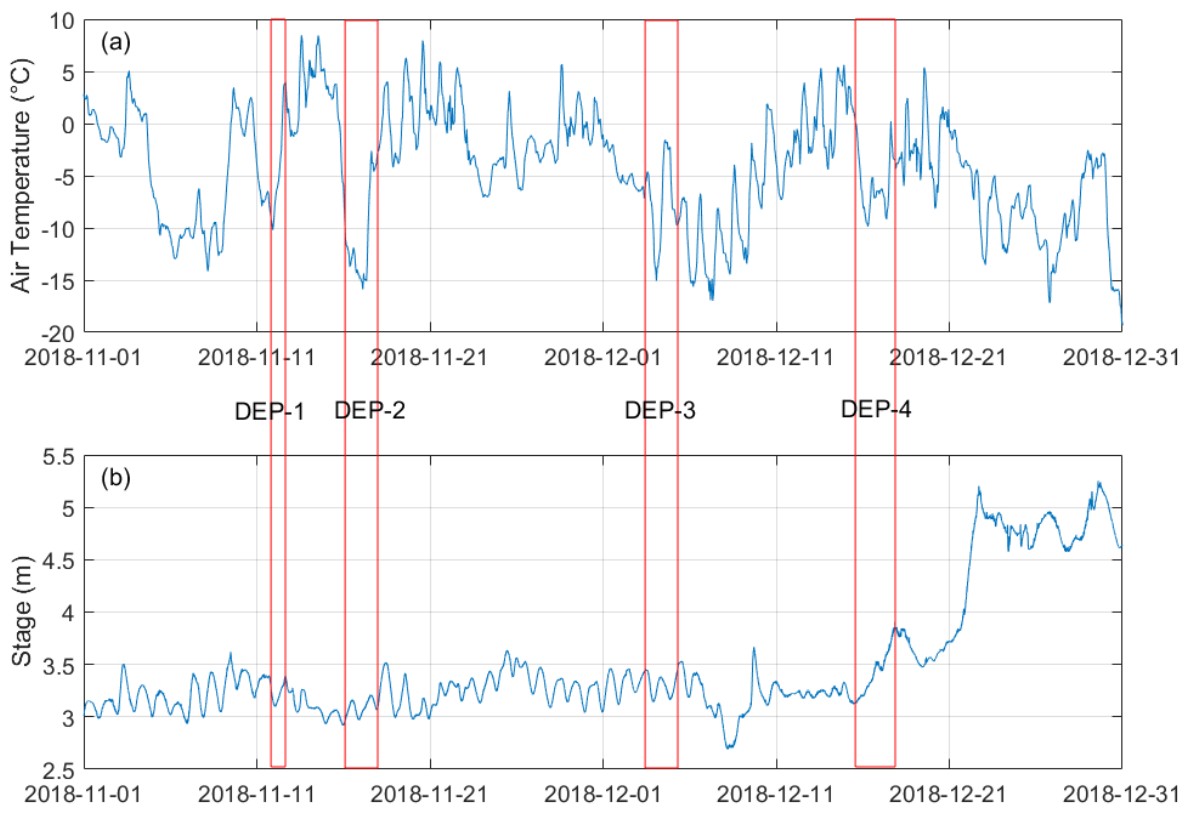

**Figure 4: Time series of (a) air temperature and (b) river stage during the 2018 freeze-up season. Field deployments are delineated with red lines.**





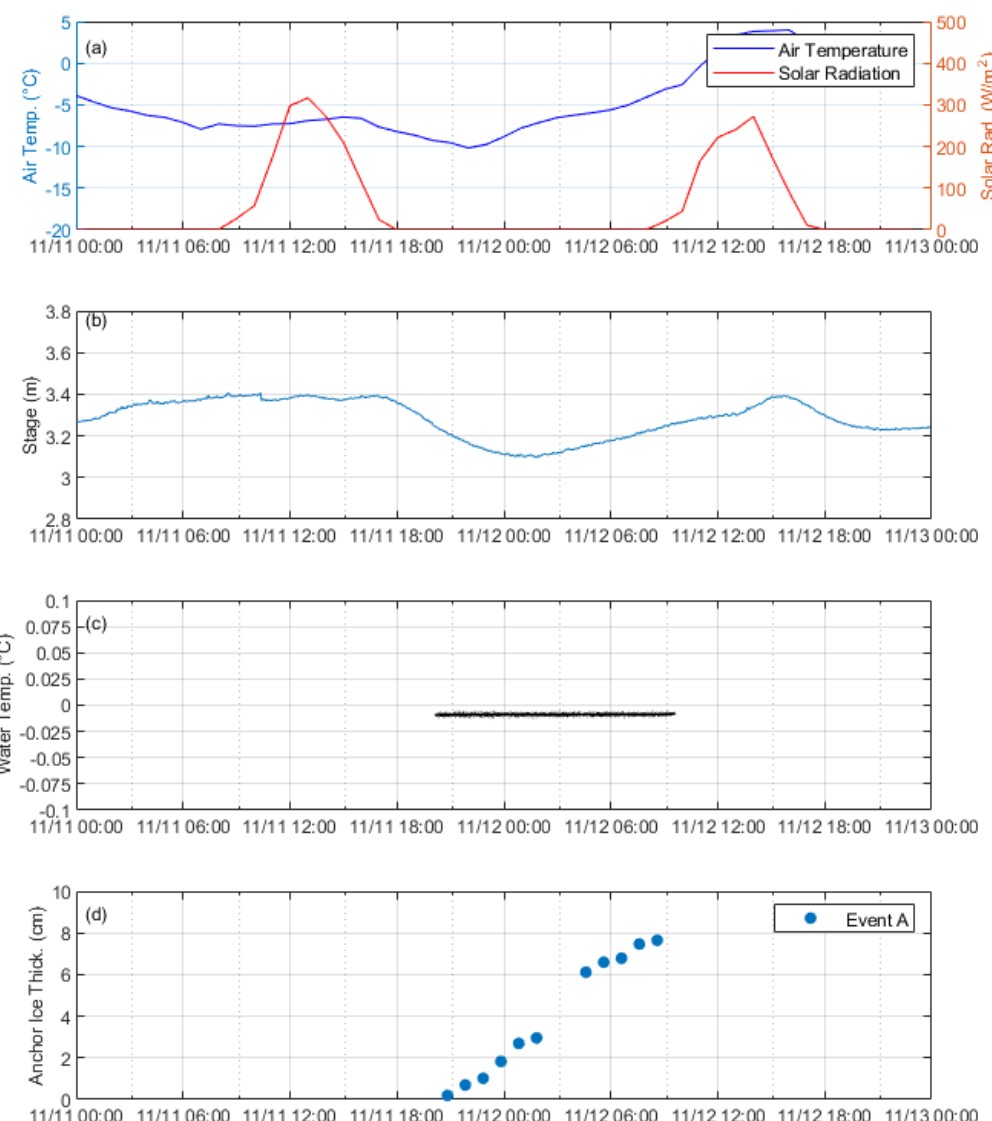

**Figure 5: Time series of results during deployment DEP-1 showing: (a) air temperature and incoming solar radiation, (b) water depth at the WSC gauge #05DF001 (c) water temperature on the substrate, and (d) anchor ice thickness above the substrate for Event A.**





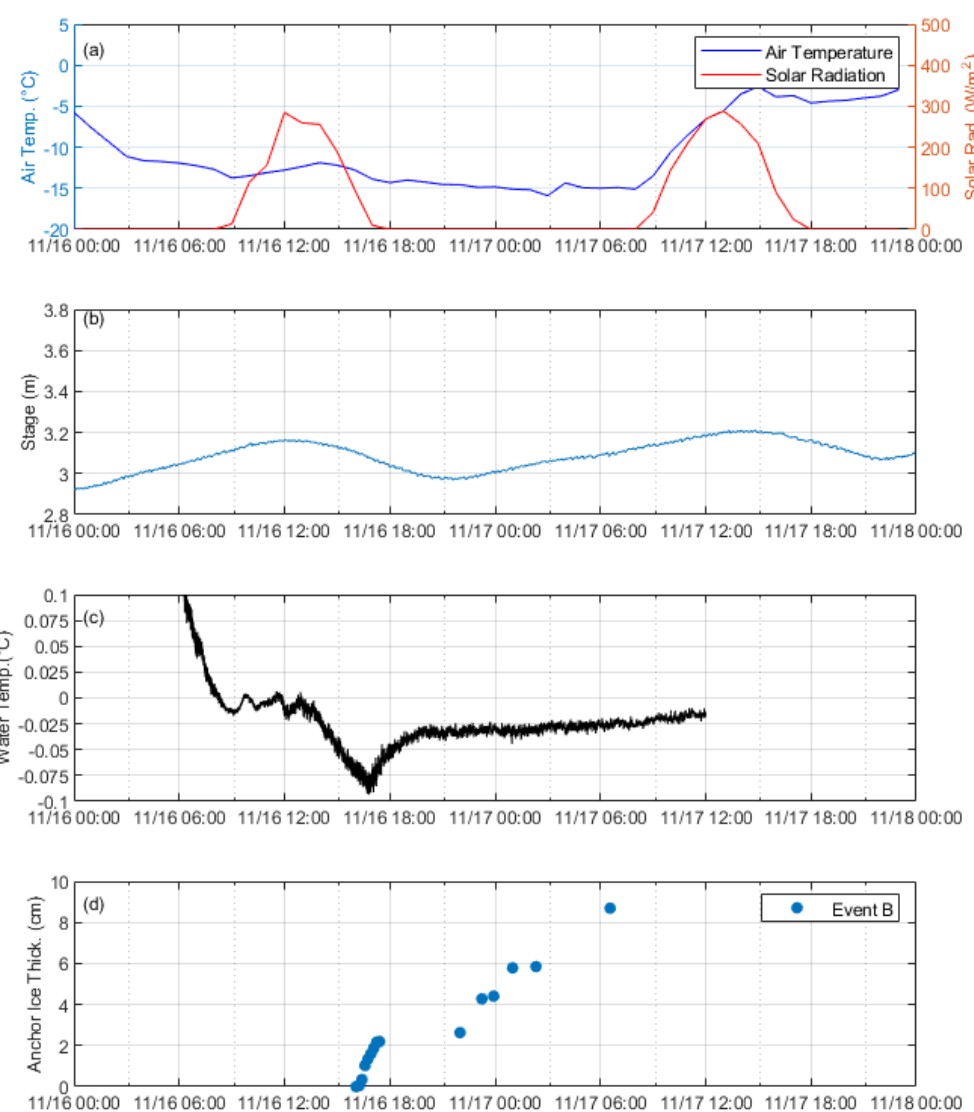

**Figure 6: Time series of results of DEP-2 showing: (a) air temperature and incoming solar radiation, (b) water depth at the WSC gauge #05DF001 (c) water temperature on the substrate, and (d) anchor ice thickness above the substrate for Event B.**

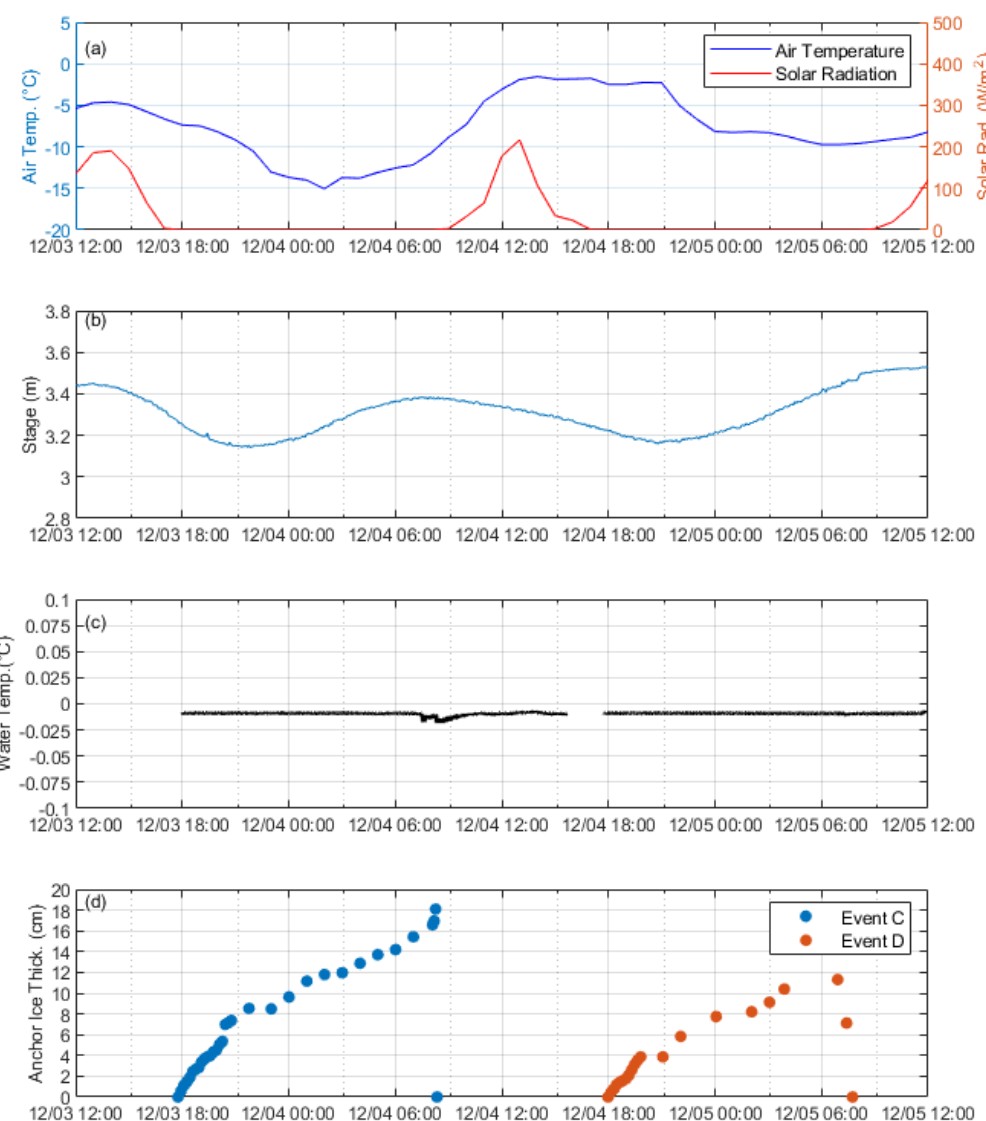

**Figure 7: Time series of results of DEP-3 showing: (a) air temperature and incoming solar radiation, (b) water depth at the WSC gauge #05DF001 (c) water temperature on the substrate, and (d) anchor ice thickness above the substrate for Events C and D.**



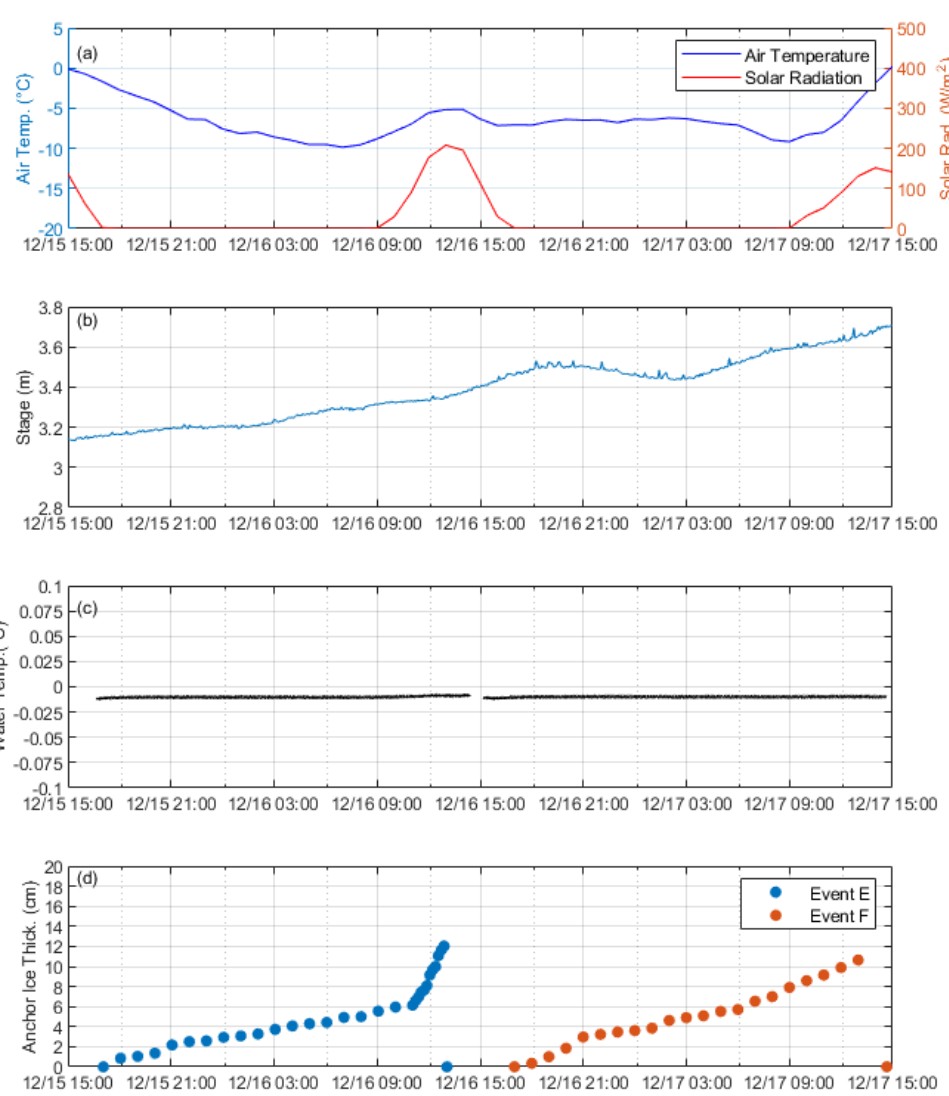

**Figure 8: Time series of results of DEP-4 showing: (a) air temperature and incoming solar radiation, (b) water depth at the WSC gauge #05DF001 (c) water temperature on the substrate, and (d) anchor ice thickness above the substrate for Events E and F.**





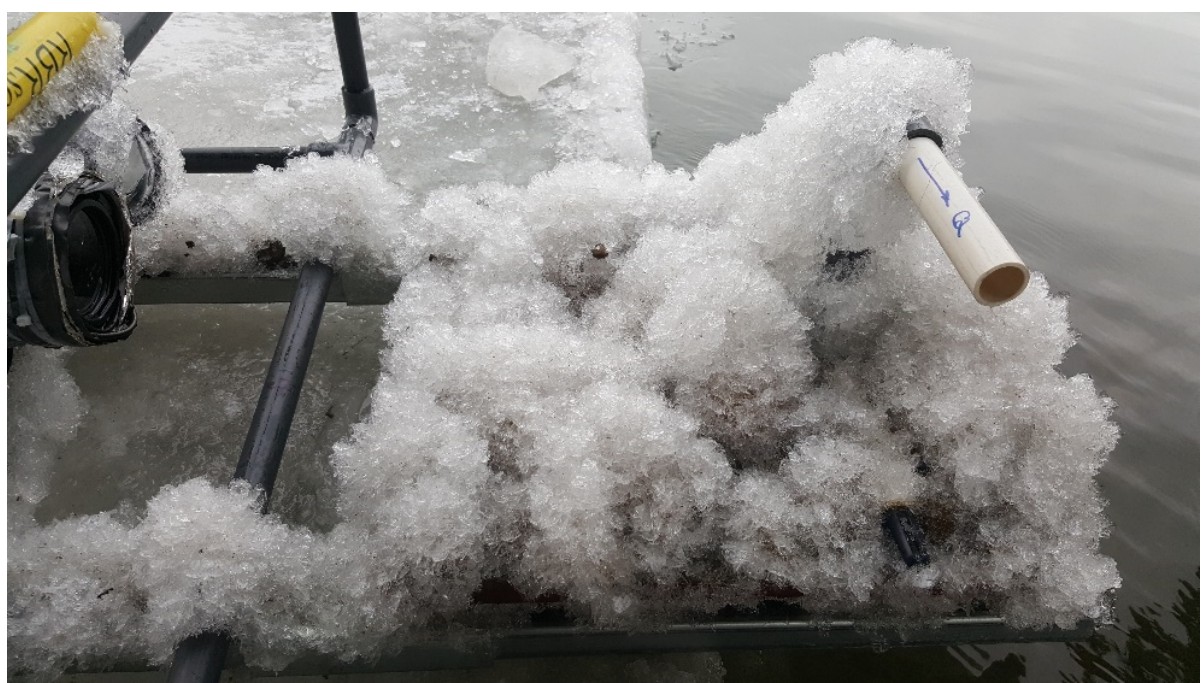

**Figure 9: Anchor ice formation on the artificial substrate after the instrument's retrieval on 12-Nov-2018.**

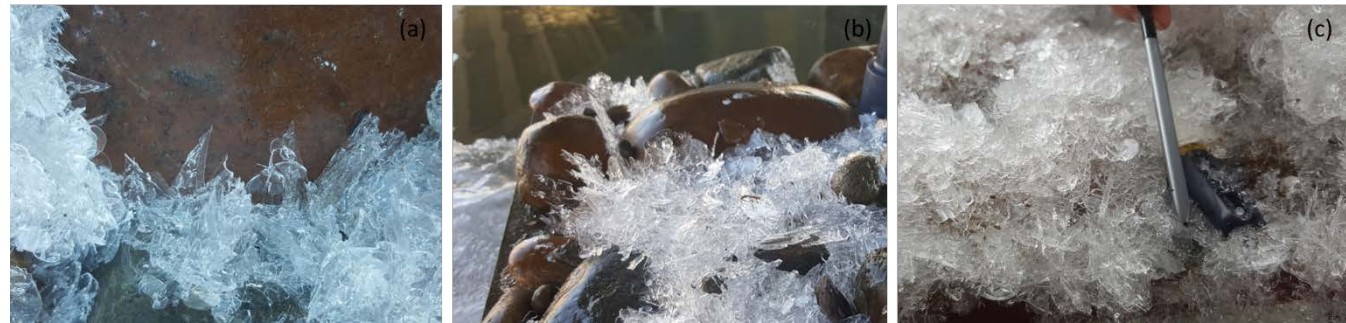

5       **Figure 10: Observed anchor ice crystal types from the 2018 freeze-up season showing (a) curved needle crystals, (b) platelet crystals, and (c) disk crystals.**




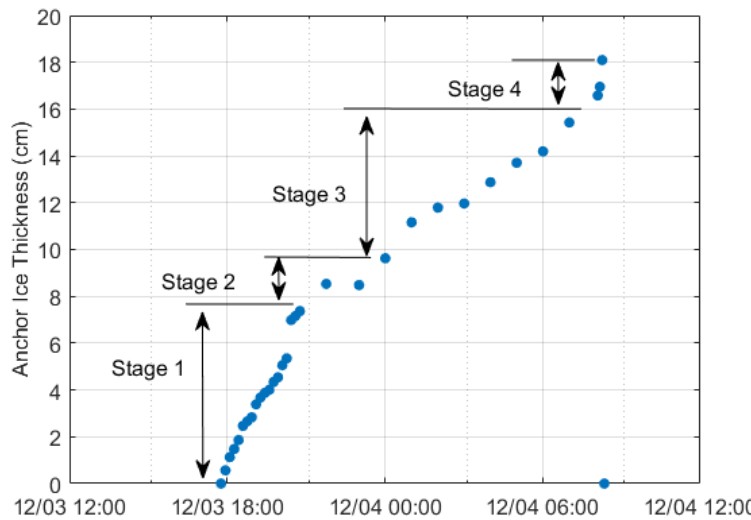

**Figure 11: Time series plot of the measured anchor ice thickness during Event C on 3-4 Dec 2018 labelled with the different stages of anchor ice formation, growth and release.**

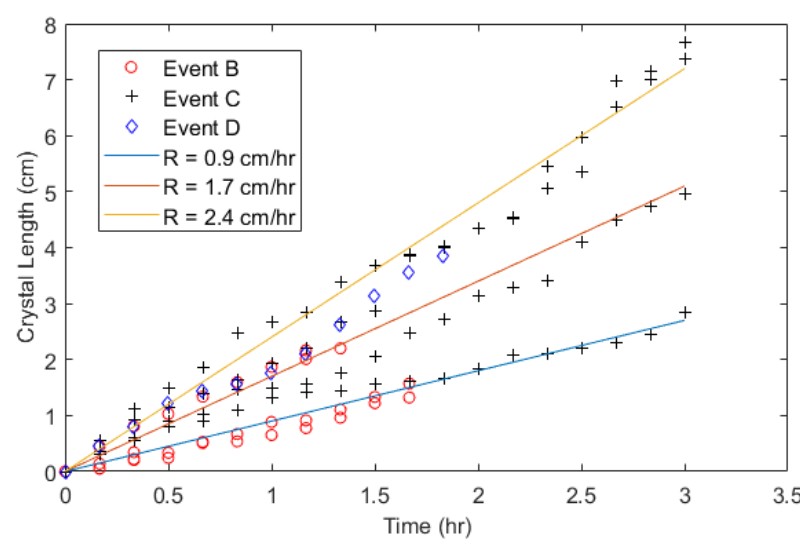

**Figure 12: Time series of individual crystal growth (Stage 1) measured from Events B, C, and D. For illustration purposes, linear growth rates, R = 0.9, 1.7, and 2.4 cm/hr were added to the plot.**

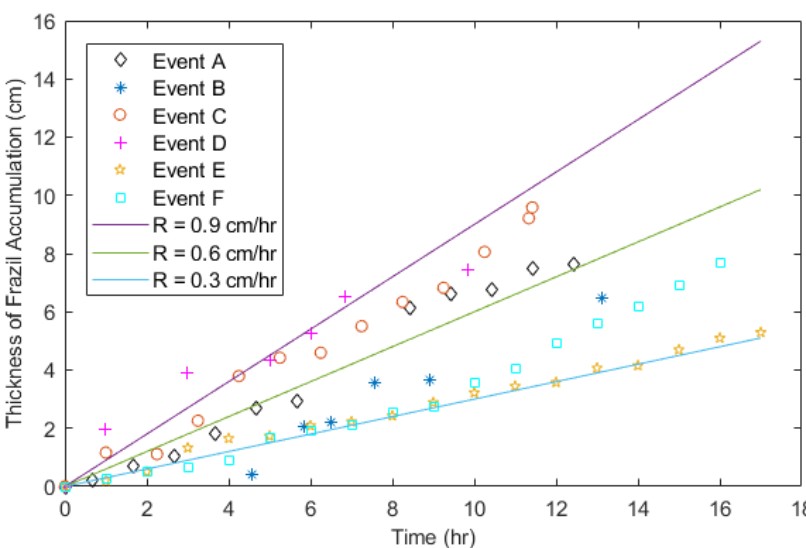

**Figure 13: Time series of anchor ice thickness growth due to frazil deposition (Stage 2 and 3) for all the measured events. For illustration purposes, linear growth rates, R = 0.3, 0.6, and 0.9 cm/hr were added to the plot.**

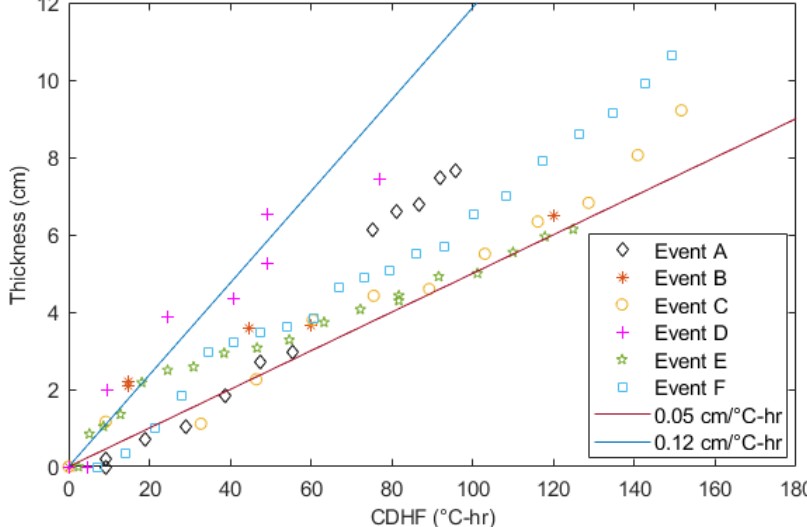

**Figure 14: Anchor ice thickness growth due to frazil deposition (Stage 2 and 3) against the cumulative degree hour of freezing, CDHF. The linear rates of 0.05 and 0.12 cm/°C-hr were added to the plot for comparison.**

