# Peer review of "Continuous in situ measurements of anchor ice formation, growth and release"

_The Cryosphere, 2020_

## Referee Comment (RC1) · Anonymous Referee #1 · 16 Aug 2020

Review of «Continuous in situ measurements of anchor ice formation, growth and release»

This is a very interesting paper addressing a topic were little data is currently available. The paper both present a novel method of sampling data and presents new insights into anchor ice growth. The paper also provides a good overview of the current knowledge of anchor ice formation and growth through the introduction. I therefore think this manuscript should be accepted for publication with some minor clarifications.

- The substrate studied was mounted quite close to the camera and camera frame. Could the camera frame have any impact on the flow field and thereby anchor ice deposits on the substrate plate.

- How would you consider the uncertainty in manually detecting the number of crystal and crystal size given the turbid/dark nature of the example picture?

- Did you compare the anchor ice forming on the substrate plate with anchor ice deposits on the natural substrate nearby the study site?

- You see anchor ice releases when the water is still supercooled and even when temperature decreases. This is an interesting observation and different from what we have observed in our anchor ice studies. What mechanism caused this? Forces from the water flow? On page 14 from line 30 you discuss the effect of rising stage e.g. from hydropeaking so this might be an explanation why ice released when the supercooling was still quite high.

- You define four stages of formation of growth. E.g. stage 2 did not appear in all experiments. Is this because this stage is not detectable or because the formation did not pass through this stage? Can you say something more on that, and do you think these four stages appear at all anchor ice formation events?

- Did you observe any difference in water temperature between the two sensors on the submerged system? I assume the temperature used is measured with the sensor closest to the substrate?

- Page13, line 10-25. Could the thick layers of anchor ice under the border ice be driven by a larger accumulation of drifting frazil? Was the structure of the deep depositions similar to the anchor ice detected at the study site? This accumulation of ice with a foundation on anchor ice is also often seen in steeper streams and where anchor ice dams form.

- Page 14, line 5-10: Did you try to estimate the heat flux during the experiments? Do you have any indication of heat transfer from the sediments? This is an interesting observation, see also comment above.

- Page 15, from line 5: This section is not very clear to me. Are you looking at providing better parametrization for modelling? I think this could have been made clearer. As

asked above, can you estimate the heat flux for your site based on climate data to test the assumptions made in the computation?

- Figure 5 – 8: I can see the reason for having the same scale for Water temp for all figures, but this obscure small variations in some of the graphs. Maybe this scale could vary between graphs?

- Figure 12: What causes the large scatter in the growth rates for event C? This is not discussed in the text where it seems like the growth followed the linear models, which in figure 12 is reasonable for B/D but not for C.

---

## Referee Comment (RC2) · Anonymous Referee #2 · 23 Aug 2020

This is an interesting paper presenting some novel finds on anchor ice formation. This paper should have a large impact on the readership. Some minor revisions are required before it is published: Page 1, Line 24: rapid what? Page 4, Line 9: should read "ice accumulation densities" Page 7, Line 8: was the flat side facing down or up on the plywood base? Page 7, Line 20: What did the change in camera housing do? Page 10, Line 30: should read "ice crystal growth" Page 10, Line 33: should read "grew at approximately linear rates . . ." Page 12. Line 9: should read: "crystals, or". Page 13, Line 13; should read "The crystal sizes" Page 15, Line 12: should read: "to be predicted at each"

---

## Referee Comment (RC3) · Anonymous Referee #3 · 25 Aug 2020

General Comments:

The authors have conducted a novel field study of anchor ice formation, growth and release that clearly addresses a knowledge gap in the literature. They have provided a thorough review of the current state of knowledge of anchor ice processes and have complemented this very well with their new field measurements. The paper is well written and will be of interest to many people in the river ice engineering field.

The authors note that several previous investigators have discussed correlations between anchor ice characteristics and the flow Froude or Reynolds number. Perhaps it would be useful to report these two values for each of the events summarized in Table

2.

There is discussion on the uncertainty of measured parameters (ie. crystal growth rate, thickness). Perhaps this could be added. I suspect that while the camera resolution (ie. pixel size) might suggest a high degree of accuracy, the problems associated with the depth of field when measuring the anchor ice thickness might be more significant.

I am unfamiliar with the local hydraulics, so I am unable to differentiate between the normal diurnal water level variations because of upstream hydropeaking versus staging during anchor ice formation and de-staging during anchor ice release. Were the impacts of anchor ice of sufficient magnitude to be observable in the water level measurements?

Specific Comments:

Line 2 of the abstract – even though line 1 notes both turbulence and supercooling, line 2 starts over and says that supercooled water generates frazil ice and does not mention the concurrent requirement of sufficient fluid turbulence.

Stage 4 is listed as the release phase (Pg 10, line 25) however, on Figure 11 Stage 4 is shown to have a very rapid increase in thickness, as opposed to a drop in thickness down to zero.

Page 12, line 25 – Did Kempema and Ettema report observed water temperature measurements at the wedge wire screens? If the water was more supercooled this could also explain the higher growth rate.

Page 14, line 1 – The substrate thermometer looked to be covered in anchor ice in the photo, which may prevent it from providing an accurate measure of the water temperature. Did you compare with the thermometer mounted higher up on the frame?

Page 16, line 6: is the first sentence too general? You've listed a few field measurements of anchor ice growth in your lit review section.

Page 16, line 9: the mode name 'rapid' could be more descriptive in my opinion.

Technical Corrections:

Pg. 1, Line 14 – rewording is required: '. . . have been reported to study'; 'but'

Pg. 1, Line 26 – repetitive 'defined'

Pg. 3, Line 23 – increase with increasing Froude number; lines 24 – 26 – changing from present to past tense a couple of times. This also occurs at other locations within the manuscript.

Pg. 3, line28 – 'have been reported' rather than 'has been reported'.

Pg. 4, line 2 – reword '. . .provided many valuable information'.

Pg. 4, line 34 – reword '. . .crystals showed grew preferentially. . .'

Pg. 14, line 33 - . . . release of event C anchor ice

―――――――――――――――――

---

## Referee Comment (RC4) · Edward Kempema (Referee) · 3 Sep 2020

I think a reasonable argument could be made that our understanding of anchor ice initiation and growth have not advanced significantly since Altberg (1936) published his findings 84 years ago. We have a particularly poor understanding of the relative importance of frazil accretion versus in situ ice growth in accumulating anchor ice masses (something Altberg struggled with). The paper by Ghobrial and Loewen describes their technique of using a high-resolution camera package to measure the growth of anchor ice (both individual ice crystals and anchor ice accumulations) on a natural cobble substrate in the North Saskatchewan River. Their paper presents preliminary data on frazil

accumulation versus in situ anchor ice growth mechanisms based on their imaging system. This paper makes a significant contribution to the ice community's understanding of anchor ice formation in this natural setting.

In my opinion the authors give short shrift to Kempema and Ettema (2013, 2016; the 2016 paper is an expanded version of the 2013 conference proceedings). These two papers describe the use of a high-resolution camera system to determine anchor-ice crystal growth rates on a wedge wire screen element placed in the Laramie River during the 2012-2013 winter. They were able to document the growth rate of individual anchor-ice ice crystals in anchor-ice masses with this system, which was similar in concept to the camera system described by Ghobrial and Loewen. Although the camera systems in both studies were similar, the two systems differed in two important ways: (1) Kempema and Ettema focused on anchor ice growth on an intake screen while the present paper focus on anchor ice on the bed; and (2) Ghobrial's and Lowen's system is much more advanced that that used by Kempema and Ettema. Specifically, the Ghobrial and Loewen system includes precise water temperature measurements to relate ice growth to supercooling levals and their camera system included a fixed, consistent cobble bed, a better camera, and heating elements to keep ice off the camera lens. This system is a major advance over what is described in Kempema and Ettema (2013, 2016) This made it possible for the authors to measure the increase of anchor-ice mass thickness in addition to measuring individual ice crystal growth. Ghobrial and Loewen reference Kempema and Ettema (2013) but appear to dismiss their reported ice crystal growth rates of ~1-4 cm/hr on the basis that the wedge wire screen was placed in the water column where heat transfer was greater relative to the bed. They suggest this might explain the higher ice-crystal growth rates reported by Kempema & Ettema relative to their findings (1-3 cm/hr) (P12L24-29, P: page number, L: line number). Considering the paucity of attached anchor ice crystal growth rates in natural settings reported in the literature (39 to my knowledge) it seems curious to dismiss ~3/4 of the observations on the basis that they were taken at the wrong point in the water column. While acknowledging that the goals of the two projects were somewhat different and the substrates were very different, my bias is that anchor ice is anchor ice, regardless of the substrate it forms on. Kempema and Ettema (2013, 2016) make the case that "frazil ice blockages" are, in fact, anchor ice. I agree with Ghobrial and Loewen that underwater ice crystal growth rates are determined by turbulent heat transfer (Altberg also concurred), but point out that every growing underwater ice crystal is in a unique, local turbulent/heat transfer environment and so will have a unique morphology representing its growth history. This reinforces my argument for including, not downplaying (dismissing?), the Kempema and Ettema (2013, 2016) ice crystal growth rates. Considering the different settings (river morphologies, water depths, weather conditions, and substrates), the observed range of growth rates are very consistent. A very real contribution of the Ghobrial and Loewen paper is that it describes a system (camera, processing software, temperature recorders, consistent bed strucure) that can be used to more (and more detailed) observations in the future.

Ghobrial and Lowen describe observations of anchor ice masses breaking the surface of the water at 1.6 m water depth near their deployment site (p13L18-25). Using their measured ice growth rates, they calculate it would take 267 hours to grow this accumulation of anchor ice. Their Figure 4 shows a 10-day period when conditions appear to have been conducive to a multi-day anchor ice cycle that could have produced this amount of ice at the rates reported in this paper. Unfortunately, the authors do not report the date of their observation. Alternatively, anchor ice growth rates may have been higher in the deeper water or released anchor ice masses (possibly negatively buoyant) were stacked one on top of the other to this thickness. The use of an average growth rate to calculate a growth time implies a much greater confidence in the average than is warranted. A possible example of released anchor ice stacking can be seen at 2:08 to 2:11 (minutes:seconds) in the manuscript video (clock time December 4, 2018 05:16 to 05:41). A frazil floc or released anchor ice mass appears on to the left of the PVC pipe in the image frame at the start of this sequence and disappears at the end. Similar processes, with potentially much larger ice masses, could have built the observed 1.6 m thick accumulation in a relatively short time. In my opinion, it

would be good to discuss these other methods of building up thick layers of anchor ice (if one can call accumulations of released ice that) rather than present the calculation. In this same section, the authors state that several auger holes showed anchor ice in contact with the underside of border ice in 1.5 m water depth. It is very common for released anchor ice to be advected under border ice in my experience. I would argue that the observation of large ice crystals has no relevance vis a vis local anchor ice growth or formation. This gets to be something of a semantic argument. Once anchor ice is released from the bed it is no longer, strictly speaking, anchor ice. By extension accumulations of this released anchor ice (slush ice?) under border ice are no longer anchor ice unless they are attached, as opposed to in contact with, the bed.

The three panels in Figure 10 purport to show (a) curved needle crystals, (b) platelet ice, and (c) ice disks. However, (a) also contains ice disks (I would call them modified frazil crystals) on the left and what looks like platelet ice on the right side of the figure; (b) does look like platelet ice; and (c) contains at least as much platelet ice as disk ice. Perhaps you could put an arrow in each panel to identify the ice crystal morphology they are meant to show? I actually think these are wonderful images, because they show the complexity that is common in an anchor ice mass (also shown in Figure 9). My experience is that most anchor ice consists of a mix of ice crystal morphologies that represent their past growth history. These photos show this wonderfully. This paper made me rethink my own concepts on anchor ice formation, which made it a pleasure to read. The paper represents a significant contribution to anchor ice research and the expectation that this technique will produce more insights on anchor ice growth mechanisms in the future.

References

Altberg, W.J. 1936, Twenty years of work in the domain of underwater ice formation, International Union of Geodesy and Geophysics, International Association of Scientific Hydrology, Bulletin 23 373-407.

Kempema Edward, W. and Ettema, R. 2016, Fish, Ice, and Wedge-Wire Screen Water Intakes, Journal of Cold Regions Engineering, 30-2, doi:10.1061/(ASCE)CR.1943-5495.0000097.

Kempema, E.W. and Ettema, R. 2013. Anchor ice and wedge-wire screens, CRIPE 17th Workshop on River Ice Processes and the Environment. CGU HS Committee on River Ice Processes and the Environment, Edmonton, Alberta, Canada, http://www.cripe.ca/docs/proceedings/17/Kempema-Ettema-2013.pdf, 15 pages.

Technical comments:

P1L7: suggest changing "cooled to slightly below 0oC" to "cooled to slightly below the freezing point" to make the definition of supercooling clear (e.g. ocean water at -1 oC is not supercooled). P3L4: Add "and" before "for collecting" P3L8: change "crystals layers" to "crystal layers" P4L34: "the crystals showed grew preferentially perpendicular to the flow" remove "showed"? P5L4: "1,800 m above sea level" not sure what this refers to. Is it the highest peak in the drainage (seems unlikely), the average elevation of the upper drainage, or what? P5,L5-10: What size classification scheme did you use? Wentworth's size classification lists sediment used in this study in the cobble size range. Gravel is not used in Wentworth's classification and boulders are >256 mm in diameter. P15L18-19: "Newly formed anchor ice accumulations likely have higher porosities because they often do not maintain their structural integrity when sampling is attempted." Is this based on personal observation or a literature reference? If this is your observation, it seems a little odd that it shows up in the discussion. At least, please, make the source clear.

---

## Author Comment (AC1) · 10 Oct 2020

Authors Response to **Referee #1** (received and published: 16 Aug 2020)

The authors wish to thank Referee #1 for the constructive comments and corrections to the discussion paper. We have responded to each of the comments from the reviewer. The comments from the reviewer are in black font and our responses are in red font.

1. Referee #1:
   This is a very interesting paper addressing a topic were little data is currently available. The paper both present a novel method of sampling data and presents new insights into anchor ice growth. The paper also provides a good overview of the current knowledge of anchor ice formation and growth through the introduction. I therefore think this manuscript should be accepted for publication with some minor clarifications.
   Authors Response:
   Thank you for your positive feedback on our paper and for highlighting the significance of the presented results.

2. Referee #1:
   The substrate studied was mounted quite close to the camera and camera frame. Could the camera frame have any impact on the flow field and thereby anchor ice deposits on the substrate plate.
   Authors Response:
   We do not think that the camera frame had a significant effect on the flow for the following reasons:
   - The camera and the frame were purposely positioned perpendicular to the flow, so that the substrate would not be inside its wake.
   - The frame was built out of 2" PVC pipe forming a hollow rectangular prism, which allowed the water to flow freely through the frame.
   - Finally, there was a 20 cm gap between the front edge of the camera and the edge of the substrate, which also helped to minimize the effect of the wake and local turbulence that would have been formed around the front vertical frame post.

3. Referee #1:
   How would you consider the uncertainty in manually detecting the number of crystal and crystal size given the turbid/dark nature of the example picture?
   Authors Response:
   The two sources of uncertainty in our results come from the accuracy of the scaling factor within each image, and the precision in detecting the same crystal in consecutive images. For the latter, we explored the feasibility of using thresholding image processing algorithms to detect and track individual particles, but this technique needed to be

calibrated and validated with sample data. Given the relatively reasonable number of sample images, we opted to manually select and track individual crystals. To do this, we printed and overlapped each pair of consecutive images (after applying a percentage of transparency to the image in MATLAB) to confirm the same crystal was identified throughout the series of images. We do have high confidence in this procedure when identifying individual crystals (Stage 1). For Stage 2, frazil deposition, the presence of a relatively high concentration of frazil crystals in the flow as well as higher turbidity levels, increased the uncertainty in detecting the top edge of the in-focus frazil deposition. See also our response number 3 to RC3 for a more quantitative description of the level of uncertainty in our scaling factor. We will add a section discussing the sources of uncertainty to the discussion section.

4. **Referee #1:**
Did you compare the anchor ice forming on the substrate plate with anchor ice deposits on the natural substrate nearby the study site?
**Authors Response:**
During each deployment we frequently visited the site to maintain the platform as well as to take pictures of sampled anchor ice depositing on the natural bed. Although we did observe anchor ice forming nearby on the bed during each event, we did not compare the characteristics of the anchor ice deposits forming on the river bed and the constructed substrate. A thorough comparison would likely have required the collection of anchor ice samples and some additional measurements which was outside of the scope of this study.

5. **Referee #1:**
You see anchor ice releases when the water is still supercooled and even when temperature decreases. This is an interesting observation and different from what we have observed in our anchor ice studies. What mechanism caused this? Forces from the water flow? On page 14 from line 30 you discuss the effect of rising stage e.g. from hydropeaking so this might be an explanation why ice released when the supercooling was still quite high.
**Authors Response:**
Thank you for highlighting this phenomenon. We did see a trend in the four release events (Events C to F) of release occurring when water levels were rising or were approaching the daily maximum (Page 15, line 3). This may indicate that hydrodynamic forces played a role in the release of these anchor ice accumulations. This is something we plan to investigate in more detail in the future.

6. **Referee #1:**
You define four stages of formation of growth. E.g. stage 2 did not appear in all experiments. Is this because this stage is not detectable or because the formation did not pass through this stage? Can you say something more on that, and do you think these four stages appear at all anchor ice formation events?
**Authors Response:**

This is a very important point. Stage 2 was defined as the transition between the rapid crystal growth stage (Stage 1), and the slower "linear" growth by frazil deposition stage (Stage 3). So, by definition, Stage 2 would possibly be observed whenever the anchor ice event was initiated by crystal growth (Stage 1). In our data, we did observe both scenarios. Page 11, line 29-31 reads:" Three of the six anchor ice events (Events B, C, and D) were observed to be initiated by in situ crystal growth (Stage 1) followed by frazil deposition. For the remaining three events (Events A, E, and F) no in situ crystal growth was observed and it appeared that the accumulations grew only by frazil deposition (Stage 2 and 3)". More research is needed to identify under which conditions, we would expect to "see" which mode of initiation.

7. **Referee #1:**
Did you observe any difference in water temperature between the two sensors on the submerged system? I assume the temperature used is measured with the sensor closest to the substrate.
**Authors Response:**
Our data showed that water temperature measurements from both sensors were almost identical within the stated accuracy of the sensors. Therefore, we decided to only show the data from the sensor on the substrate since it is closer to the anchor ice formation. We will add this note to the manuscript when introducing these results in Page 9, line 5.

8. **Referee #1:**
Page13, line 10-25. Could the thick layers of anchor ice under the border ice be driven by a larger accumulation of drifting frazil? Was the structure of the deep depositions similar to the anchor ice detected at the study site? This accumulation of ice with a foundation on anchor ice is also often seen in steeper streams and where anchor ice dams form.
**Authors Response:**
This is an interesting comment. We agree that the thick "anchor ice" observed under the border ice may not be due entirely to growth of locally forming anchor ice. We did not collect samples of these deep deposits, so we do not know if the structure was similar to open water anchor ice formations. A comment noting that "the sources of those thick deposits may be due to local anchor ice growth, or accumulation of floating frazil slush or stacking of released anchor ice from upstream, or a combination of any of these phenomenon" will be added to the revised paper.

9. **Referee #1:**
Page 14, line 5-10: Did you try to estimate the heat flux during the experiments? Do you have any indication of heat transfer from the sediments? This is an interesting observation, see also comment above.
**Authors Response:**
In this study we did not estimate each heat flux component during each event, but we qualitatively discussed the expected effects of meteorological forcing on the release of

anchor ice (Page 14, line 5-10). In addition, we estimated the maximum net heat loss from the water to be 300 W/m$^2$ using a liner heat transfer equation (Page 16, line 2). We did not conduct direct measurements of water temperatures in the riverbed that would be required to estimate heat transfer from the sediment, but we thought it was worthwhile to list it as a potential source of heat input that might weaken the bond between anchor ice and the substrate (Page 14, line 4).

10. **Referee #1:**
Page 15, from line 5: This section is not very clear to me. Are you looking at providing better parametrization for modelling? I think this could have been made clearer. As asked above, can you estimate the heat flux for your site based on climate data to test the assumptions made in the computation?
**Authors Response:**
We did estimate a maximum net heat flux between air and water using the linear heat transfer equation to be 300 W/m$^2$ (Page 16, line 2). The objective of this section of the discussion was to use the measured rates of growth to provide a realistic range of values for suspended frazil concentration and the porosity of anchor ice using Equation 1. This is discussed in page 15 and 16. We will review and if necessary, rewrite parts of this section to improve its clarity and better explain the rationale.

11. **Referee #1:**
Figure 5 – 8: I can see the reason for having the same scale for Water temp for all figures, but this obscure small variations in some of the graphs. Maybe this scale could vary between graphs?
**Authors Response:**
We think it is easier for the reader to compare between events when we use the same scale. For most of the events the small variations in the water temperatures are insignificant and within the stated sensors accuracy of ±0.002 °C.

12. **Referee #1:**
Figure 12: What causes the large scatter in the growth rates for event C? This is not discussed in the text where it seems like the growth followed the linear models, which in figure 12 is reasonable for B/D but not for C.?
**Authors Response:**
This is a very valid comment. We do not know the reason for the scatter in Event C. This scatter shows that crystals can grow at significantly different rates during the same time interval and in close proximity to each other. This might be because crystals can originate from different parts of the substrate and as a result, they will be exposed to different flow conditions. We will add a discussion in Page 12 line 19-29, to highlight this phenomenon.

---

## Author Comment (AC2) · 10 Oct 2020

Authors Response to **Referee #2** (received and published: 23 August 2020)

The authors wish to thank Referee #2 for their constructive comments and suggestions. We have responded to each of the comments from the reviewer. The comments from the reviewer are in black font and our responses are in red font.

1. **Referee #2:**
   This is an interesting paper presenting some novel finds on anchor ice formation. This paper should have a large impact on the readership. Some minor revisions are required before it is published.
   **Authors Response:**
   Thank you for your positive feedback on our paper and for highlighting the significance of the presented results.

2. **Referee #2:**
   Page 1, Line 24: rapid what?
   **Authors Response:**
   We will clarify the last sentence of the abstract to read:" Anchor ice was observed releasing from the bed in three modes referred to as lifting of the entire accumulation, shearing of layers of the accumulation and rapid release of the entire accumulation."

3. **Referee #2:**
   Page 4, Line 9: should read "ice accumulation densities"?
   **Authors Response:**
   Thank you for catching this mistake. The text will be updated as suggested.

4. **Referee #2:**
   Page 7, Line 8: was the flat side facing down or up on the plywood base?
   **Authors Response:**
   The flat side of the substrate materials were facing down on the plywood base to increase the gluing contact surface between the base and the gravel.

5. **Referee #2:**
   Page 7, Line 20: What did the change in camera housing do?
   **Authors Response:**
   The camera housing was changed due to some damage to the original housing resulting in water leaking inside the case.

6. **Referee #2:**
   Page 10, Line 30: should read "ice crystal growth"?

**Authors Response:**

Thank you for catching this mistake. The text will be updated as suggested.

7. **Referee #2:**

Page 10, Line 33: should read "grew at approximately linear rates …".

**Authors Response:**

Thank you for catching this mistake. The text will be updated as suggested.

8. **Referee #2:**

Page 12. Line 9: should read: "crystals, or".

**Authors Response:**

Thank you for catching this mistake. The text will be updated as suggested.

9. **Referee #2:**

Page 13, Line 13; should read "The crystal sizes".

**Authors Response:**

Thank you for catching this mistake. The text will be updated as suggested.

10. **Referee #2:**

Page 15, Line 12: should read: "to be predicted at each"?

**Authors Response:**

Thank you for catching this mistake. The text will be updated as suggested.

---

## Author Comment (AC3) · 10 Oct 2020

Authors Response to **Referee #3** (received and published: 25 August 2020)

The authors wish to thank Referee #3 for the constructive comments and suggested corrections to the discussion paper. We have responded to each of the comments from the reviewer. The comments from the reviewer are in black font and our responses are in red font.

1. **Referee #3:**
   The authors have conducted a novel field study of anchor ice formation, growth and release that clearly addresses a knowledge gap in the literature. They have provided a thorough review of the current state of knowledge of anchor ice processes and have complemented this very well with their new field measurements. The paper is well written and will be of interest to many people in the river ice engineering field.
   **Authors Response:**
   Thank you for your positive feedback on our paper and for highlighting the significance of the presented results.

2. **Referee #3:**
   The authors note that several previous investigators have discussed correlations between anchor ice characteristics and the flow Froude or Reynolds number. Perhaps it would be useful to report these two values for each of the events summarized in Table 2.
   **Authors Response:**
   This is a very useful suggestion. Unfortunately, during these measurements, we did not measure local velocities and depths. We did make measurements of these flow parameters in the following year of measurements and will include them in future publications.

3. **Referee #3:**
   There is discussion on the uncertainty of measured parameters (ie. crystal growth rate, thickness). Perhaps this could be added. I suspect that while the camera resolution (ie. pixel size) might suggest a high degree of accuracy, the problems associated with the depth of field when measuring the anchor ice thickness might be more significant.
   **Authors Response:**
   We agree that a discussion on the sources of uncertainty is needed. This comment was also raised by Referee #1. We identified three the sources of uncertainty: (1) the camera resolution, (2) the precision in detecting the same crystal between consecutive images, and (3) image clarity when trying to identify the in-focus anchor ice that we tracked to measure growth rates. We agree that the high camera resolution only indicates the maximum possible accuracy and that other factors will govern the actual measurement accuracy. Images clarity did affect the uncertainty in tracking individual crystals and top of ice accumulation. When tracking individual crystals, we printed and overlapped each

pair of consecutive images (after applying a percentage of transparency to the image in MATLAB) to confirm that the same crystal was identified throughout the series of images. We do have high confidence in this procedure when identifying individual crystals (Stage 1). For Stage 2, frazil deposition, high concentrations of frazil crystals in the flow as well as higher turbidity levels, increased the uncertainty in detecting the top edge of the in-focus frazil deposition. This leads to the third source of uncertainty the accuracy of the scaling factor. The in-focus section of the substrate used for scaling anchor ice sizes was 40 cm away from the face of the lens. We estimate that the vast majority of observed anchor ice was located between 30 to 50 cm away from the camera, ±10 cm from the focus distance. If we considered these two extreme cases (i.e. ± 10 cm), the resulting expected error in estimating anchor ice dimensions (crystals or depth of deposition) would be approximately ± 25%. We will include a section in the discussion addressing the sources of uncertainty.

4. **Referee #3:**
   I am unfamiliar with the local hydraulics, so I am unable to differentiate between the normal diurnal water level variations because of upstream hydropeaking versus staging during anchor ice formation and de-staging during anchor ice release. Were the impacts of anchor ice of sufficient magnitude to be observable in the water level measurements?
   **Authors Response:**
   Based on our knowledge of the site, we do not think that anchor ice formation and release have any significant effects on the change in water levels.  The diurnal variation of water levels appears to be entirely controlled by the hydropeaking from the dam's operation upstream. This was observed during the first three deployment. During DEP-4 (Events E and F), the continuously rising water levels were attributed to the staging-up due to ice cover formation downstream. We did discuss our assessment of the water level data in Page 14 line 28 to Page 15 line 4.

5. **Referee #3:**
   Line 2 of the abstract – even though line 1 notes both turbulence and supercooling, line 2 starts over and says that supercooled water generates frazil ice and does not mention the concurrent requirement of sufficient fluid turbulence.
   **Authors Response:**
   Thank you for your comment. Line 2 of the abstract will be updated to read:" In supercooled turbulent water…".

6. **Referee #3:**
   Stage 4 is listed as the release phase (Pg 10, line 25) however, on Figure 11 Stage 4 is shown to have a very rapid increase in thickness, as opposed to a drop in thickness down to zero.
   **Authors Response:**

Thank you for clarifying this. The label for Stage 4 in Figure 11 was to highlight the start of the "lifting" release mechanism before the total removal of the anchor ice accumulation. We will update the figure caption to clarify this issue.

7. **Referee #3:**
Page 12, line 25 – Did Kempema and Ettema report observed water temperature measurements at the wedge wire screens? If the water was more supercooled this could also explain the higher growth rate.
**Authors Response:**
Unfortunately, Kempema and Ettema did not measure water temperatures in their setup. We attributed their higher growth rates to higher flow turbulence (the wedge wire screen being installed 23 cm above the bed).

8. **Referee #3:**
Page 14, line 1 – The substrate thermometer looked to be covered in anchor ice in the photo, which may prevent it from providing an accurate measure of the water temperature. Did you compare with the thermometer mounted higher up on the frame?
**Authors Response:**
Yes, we did compare results from both sensors and the data showed that water temperature measurements from both sensors were almost identical within the stated accuracy of the sensors. Therefore, we decided to only show the data from the sensor on the substrate since it is closer to the anchor ice formation.

9. **Referee #3:**
Page 16, line 6: is the first sentence too general? You've listed a few field measurements of anchor ice growth in your lit review section.
**Authors Response:**
Yes, we agree with your comment. We will update the sentence to read:" The first continuous field measurements of the anchor ice cycle including, initiation, growth and release mechanisms were captured in this study".

10. **Referee #3:**
Page 16, line 9: the mode name 'rapid' could be more descriptive in my opinion.
**Authors Response:**
The only alternative to "rapid" that we thought might be applicable was "instantaneous". However, because our images were taken every 5 min we did not think it was accurate to call this instantaneous release.

11. **Referee #3:**
Pg. 1, Line 14 – rewording is required: '… have been reported to study'; 'but'
**Authors Response:**
Agreed. The sentence will be reworded to read: "Although detailed laboratory experiments studying anchor ice have been reported in the literature, but very few field measurements of anchor ice processes have been reported.".

**12. Referee #3:**
Pg. 1, Line 26 – repetitive 'defined'
**Authors Response:**
Agreed. The sentence will be reworded to read: "Anchor ice is described as ice that is attached or "anchored" to the bed of natural water bodies (rivers, lakes or sea floors) as defined by World Meteorological Organization (1970).".

**13. Referee #3:**
Pg. 3, Line 23 – increase with increasing Froude number; lines 24 – 26 – changing from present to past tense a couple of times. This also occurs at other locations within the manuscript.
**Authors Response:**
Thank you for catching this mistake. The sentence in line 22 will read:" anchor ice growth rates and densities increased with increasing Froude number." We will also review the manuscript to verify consistency in using the correct tense.

**14. Referee #3:**
Pg. 3, line28 – 'have been reported' rather than 'has been reported'.
**Authors Response:**
Updated.

**15. Referee #3:**
Pg. 3, line28 – 'have been reported' rather than 'has been reported'.
**Authors Response:**
Updated.

**16. Referee #3:**
Pg. 4, line 2 – reword '…provided many valuable information'.
**Authors Response:**
Agreed. The sentence will read:" Despite these limitations, field studies have significantly advanced our knowledge of anchor ice processes".

**17. Referee #3:**
Pg. 4, line 34 – reword '… crystals showed grew preferentially…'.
**Authors Response:**
Agreed. The sentence will read:" In both cases, the crystals showed preferential growth perpendicular to the flow.".

**18. Referee #3:**
Pg. 14, line 33 - … release of event C anchor ice
**Authors Response:**
Thank you for catching this mistake. We will update the sentence to read:" The release of event C anchor ice coincided with a peak in the daily water levels of 3.38 m."

---

## Author Comment (AC4)

Authors Response to **Dr Edward Kempema** kempema@uwyo.edu (Referee #4) (received and published: 3 September 2020)

The authors wish to thank Dr. Kempema for the constructive comments and suggested corrections to the discussion paper. We have responded to each of his comments. The comments are in black font and our responses are in red font.

1. **Dr. Kempema:**
   I think a reasonable argument could be made that our understanding of anchor ice initiation and growth have not advanced significantly since Altberg (1936) published his findings 84 years ago. We have a particularly poor understanding of the relative importance of frazil accretion versus in situ ice growth in accumulating anchor ice masses (something Altberg struggled with). The paper by Ghobrial and Loewen describes their technique of using a high-resolution camera package to measure the growth of anchor ice (both individual ice crystals and anchor ice accumulations) on a natural cobble substrate in the North Saskatchewan River. Their paper presents preliminary data on frazil accumulation versus in situ anchor ice growth mechanisms based on their imaging system. This paper makes a significant contribution to the ice community's understanding of anchor ice formation in this natural setting.
   **Authors Response:**
   Thank you for your positive feedback on our paper and for highlighting the significance of the presented results.

2. **Dr. Kempema:**
   In my opinion the authors give short shrift to Kempema and Ettema (2013, 2016; the 2016 paper is an expanded version of the 2013 conference proceedings). These two papers describe the use of a high-resolution camera system to determine anchor-ice crystal growth rates on a wedge wire screen element placed in the Laramie River during the 2012-2013 winter. They were able to document the growth rate of individual anchor-ice ice crystals in anchor-ice masses with this system, which was similar in concept to the camera system described by Ghobrial and Loewen. Although the camera systems in both studies were similar, the two systems differed in two important ways: (1) Kempema and Ettema focused on anchor ice growth on an intake screen while the present paper focus on anchor ice on the bed; and (2) Ghobrial's and Lowen's system is much more advanced that that used by Kempema and Ettema. Specifically, the Ghobrial and Loewen system includes precise water temperature measurements to relate ice growth to supercooling levals and their camera system included a fixed, consistent cobble bed, a better camera, and heating elements to keep ice off the camera lens. This system is a major advance over what is described in Kempema and Ettema (2013, 2016) This made it possible for the authors to measure the increase of anchor-ice mass thickness in addition to measuring individual ice crystal growth. Ghobrial and Loewen

reference Kempema and Ettema (2013) but appear to dismiss their reported ice crystal growth rates of ~1-4 cm/hr on the basis that the wedge wire screen was placed in the water column where heat transfer was greater relative to the bed. They suggest this might explain the higher ice-crystal growth rates reported by Kempema & Ettema relative to their findings (1-3 cm/hr) (P12L24-29, P: page number, L: line number). Considering the paucity of attached anchor ice crystal growth rates in natural settings reported in the literature (39 to my knowledge) it seems curious to dismiss ~3/4 of the observations on the basis that they were taken at the wrong point in the water column. While acknowledging that the goals of the two projects were somewhat different and the substrates were very different, my bias is that anchor ice is anchor ice, regardless of the substrate it forms on. Kempema and Ettema (2013, 2016) make the case that "frazil ice blockages" are, in fact, anchor ice. I agree with Ghobrial and Loewen that underwater ice crystal growth rates are determined  by turbulent heat transfer (Altberg also concurred), but point out that every growing underwater ice crystal is in a unique, local turbulent/heat transfer environment and so will have a unique morphology representing its growth history. This reinforces my argument for including, not downplaying (dismissing?), the Kempema and Ettema (2013, 2016) ice crystal growth rates. Considering the different settings (river morphologies, water depths, weather conditions, and substrates), the observed range of growth rates are very consistent. A very real contribution of the Ghobrial and Loewen paper is that it describes a system (camera, processing software, temperature recorders, consistent bed strucure) that can be used to more (and more detailed) observations in the future.

**Authors Response:**
Thank you for highlighting the work of Altberg (1936) and Kempema and Ettema (2013 and 2016). We did refer to the system and the growth rates reported by Kempema and Ettema (2013) in page 12, lines 23-29. Nevertheless, we agree that it is important to include a more detailed description of their system and findings in the literature review as well as the discussion section. Also, the following references will be added to the revised paper:

- Altberg, W.J. 1936, Twenty years of work in the domain of underwater ice formation, International Union of Geodesy and Geophysics, International Association of Scientific Hydrology, Bulletin 23 373-407.
- Kempema Edward, W. and Ettema, R. 2016, Fish, Ice, and Wedge-Wire Screen Water Intakes, Journal of Cold Regions Engineering, 30-2, doi:10.1061/(ASCE)CR.1943-5495.0000097.

3. **Dr. Kempema:**
Ghobrial and Loewen describe observations of anchor ice masses breaking the surface of the water at 1.6 m water depth near their deployment site (p13L18-25). Using their measured ice growth rates, they calculate it would take 267 hours to grow this accumulation of anchor ice. Their Figure 4 shows a 10-day period when conditions appear to have been conducive to a multi-day anchor ice cycle that could have produced this amount of ice at the rates reported in this paper. Unfortunately, the authors do not report the date of their observation. Alternatively, anchor ice growth rates may have

been higher in the deeper water or released anchor ice masses (possibly negatively buoyant) were stacked one on top of the other to this thickness. The use of an average growth rate to calculate a growth time implies a much greater confidence in the average than is warranted.  A possible example of released anchor ice stacking can be seen at 2:08 to 2:11 (minutes:seconds) in the manuscript video (clock time December 4, 2018 05:16 to 05:41). A frazil floc or released anchor ice mass appears on to the left of the PVC pipe in the image frame at the start of this sequence and disappears at the end. Similar processes, with potentially much larger ice masses, could have built the observed 1.6 m thick accumulation in a relatively short time.  In my opinion, it would be good to discuss these other methods of building up thick layers of anchor ice (if one can call accumulations of released ice that) rather than present the calculation.  In this same section, the authors state that several auger holes showed anchor ice in contact with the underside of border ice in 1.5 m water depth. It is very common for released anchor ice to be advected under border ice in my experience. I would argue that the observation of large ice crystals has no relevance vis a vis local anchor ice growth or formation. This gets to be something of a semantic argument. Once anchor ice is released from the bed it is no longer, strictly speaking, anchor ice. By extension accumulations of this released anchor ice (slush ice?) under border ice are no longer anchor ice unless they are attached, as opposed to in contact with, the bed.

**Authors Response:**
We want to thank Dr Kempema for providing these descriptions of other possible sources and mechanisms of anchor ice accumulation such as the effect of stacking of released anchor ice or buildup of suspended frazil slush to existing anchor ice accumulations. As suggested, we will include a more in-depth discussion of possible explanations for the thick deposits of anchor ice that we observed and also for the thick accumulations we observed under border ice.

4. **Dr. Kempema:**
The three panels in Figure 10 purport to show (a) curved needle crystals, (b) platelet ice, and (c) ice disks. However, (a) also contains ice disks (I would call them modified frazil crystals) on the left and what looks like platelet ice on the right side of the figure; (b) does look like platelet ice; and (c) contains at least as much platelet ice as disk ice. Perhaps you could put an arrow in each panel to identify the ice crystal morphology they are meant to show? I actually think these are wonderful images, because they show the complexity that is common in an anchor ice mass (also shown in Figure 9). My experience is that most anchor ice consists of a mix of ice crystal morphologies that represent their past growth history. These photos show this wonderfully.

**Authors Response:**
Thank you for highlighting the importance of showing such images of anchor ice crystals. As suggested, we will add arrows to the photographs indicating the different crystal morphologies in each image.  We will also add a brief discussion of how these images demonstrate the complexity that is commonly observed in anchor ice accumulations.

5. **Dr. Kempema:**

This paper made me rethink my own concepts on anchor ice formation, which made it a pleasure to read. The paper represents a significant contribution to anchor ice research and the expectation that this technique will produce more insights on anchor ice growth mechanisms in the future.

**Authors Response:**

Thank you for your positive comments and encouragement.

**Technical Comments:**

6. **Dr. Kempema:**
   P1L7: suggest changing "cooled to slightly below 0oC" to "cooled to slightly below the freezing point" to make the definition of supercooling clear (e.g. ocean water at -1 oC is not supercooled).
   **Authors Response:**
   Agree.

7. **Dr. Kempema:**
   P3L4: Add "and" before "for collecting".
   **Authors Response:**
   Agree.

8. **Dr. Kempema:**
   P3L8: change "crystals layers" to "crystal layers"
   **Authors Response:**
   Agree.

9. **Dr. Kempema:**
   P4L34: "the crystals showed grew preferentially perpendicular to the flow" remove "showed"?
   **Authors Response:**
   Agree.

10. **Dr. Kempema:**
    P6L4: "1,800 m above sea level" not sure what this refers to. Is it the highest peak in the drainage (seems unlikely), the average elevation of the upper drainage, or what?
    **Authors Response:**
    This refers to the mouth of the glacier feeding into the North Saskatchewan River.

11. **Dr. Kempema:**
    P7,L5-10: What size classification scheme did you use? Wentworth's size classification lists sediment used in this study in the cobble size range. Gravel is not used in Wentworth's classification and boulders are >256 mm in diameter.
    **Authors Response:**

We performed a sieve analysis of the bed samples and used sediment particles on the substrate that ranged in size between 3.8 cm and 12.5 cm. According to the classification of naturally occurring sediments reported in the USGS Scientific Investigations Report 2019–5073, the range of sediment size used on the substrate would be classified as very coarse pebble gravel and fine cobble gravel.

12. **Dr. Kempema:**
P15L18-19: "Newly formed anchor ice accumulations likely have higher porosities because they often do not maintain their structural integrity when sampling is attempted." Is this based on personal observation or a literature reference? If this is your observation, it seems a little odd that it shows up in the discussion. At least, please, make the source clear.
**Authors Response:**
This was based on our observations and the observations of Dubé et al. (2014). We will revise the paper accordingly.